# GeSS: Benchmarking Geometric Deep Learning under Scientific Applications with Distribution Shifts

**Deyu Zou[1]***, **Shikun Liu[2]***, **Siqi Miao[2], Victor Fung[2], Shiyu Chang[3], Pan Li[2]**
[1]University of Science and Technology of China, `zoudeyu2020@mail.ustc.edu.cn`,
[2]Georgia Institute of Technology, `{shikun.liu,siqi.miao,victorfung,panli}@gatech.edu`
[3]University of California, Santa Barbara, `chang87@ucsb.edu`

## Abstract

Geometric deep learning (GDL) has gained significant attention in scientific fields, for its proficiency in modeling data with intricate geometric structures. Yet, very few works have delved into its capability of tackling the distribution shift problem, a prevalent challenge in many applications. To bridge this gap, we propose GeSS, a comprehensive benchmark designed for evaluating the performance of GDL models in scientific scenarios with distribution shifts. Our evaluation datasets cover diverse scientific domains from particle physics, materials science to biochemistry, and encapsulate a broad spectrum of distribution shifts including conditional, covariate, and concept shifts. Furthermore, we study three levels of information access from the out-of-distribution (OOD) test data, including no OOD information, only unlabeled OOD data, and OOD data with a few labels. Overall, our benchmark results in 30 different experiment settings, and evaluates 3 GDL backbones and 11 learning algorithms in each setting. A thorough analysis of the evaluation results is provided, poised to illuminate insights for GDL researchers and domain practitioners who are to use GDL in their applications.

## 1 Introduction

Machine learning (ML) techniques, as a powerful and efficient approach, have been widely used in diverse scientific fields, including high energy physics (HEP) [15], materials science [20], and drug discovery [81], propelling ML4S (ML for Science) into a promising direction. In particular, geometric deep learning (GDL) is gaining much focus in scientific applications because many scientific data can be represented as point cloud data embedded in a complex geometric space. Current GDL research mainly focuses on neural network architectures design [79, 19, 36, 73, 78, 47], capturing geometric properties (*e.g.,* invariance and equivariance properties), to learn useful representations for geometric data, and these backbones have shown to be successful in various GDL scenarios.

However, ML models in scientific applications consistently face challenges related to data distribution shifts ($\mathbb{P}_{\mathcal{S}}(X, Y) \neq \mathbb{P}_{\mathcal{T}}(X, Y)$) between the training (source) domain $\mathcal{S}$ and the test (target) domain $\mathcal{T}$. In particular, the regime expected to have new scientific discoveries has often been less explored and thus holds limited data with labels. To apply GDL techniques to such a regime, researchers often resort to training models over labeled data from well-explored regimes or theory-guided simulations, whose distribution may not align well with the real-world to-be-explored regime of scientific interest. In materials science, for example, the OC20 dataset [10] covers a broad space of catalyst surfaces and adsorbates. ML models trained over this dataset may be expected to extrapolate to new catalyst compositions such as oxide electrocatalysts [80]. Additionally, in HEP, models are often trained based on simulated data and are expected to generalize to real experiments, which hold more variable conditions and may differ substantially from simulations [50].

---

*Equal contribution. Code and data are available at `https://github.com/Graph-COM/GESS`.

Despite the significance, scant research has systematically explored the distribution shift challenges specific to GDL. Findings from earlier studies on CV and NLP tasks [9, 13, 99] might not be directly applicable to GDL models, due to the substantially distinct model architectures.

In the context of ML4S, several studies address model generalization issues, but there are two prominent *disparities* in these works. First, previous studies are often confined to specific scientific scenarios that have different types of distribution shifts. For example, [96] concentrated exclusively on drug-related shifts such as scaffold shift, while [33] investigated model generalization to deal with the label-fidelity shifts in the application of materials property prediction. Due to the disparity in shift types, the findings effective for one application might be ineffectual for another.

Moreover, existing studies often assume different levels of the availability of target-domain data information. Specifically, while some studies assume some availability of the data from the target domain [33], they differ on whether such data is labeled or not. On the other hand, certain investigations presume total unavailability of the target-domain data [55]. These varying conditions often dictate the selection of corresponding methodologies.

To address the above disparities, this paper presents **GeSS**, a benchmark to evaluate GDL models' capability of dealing with various types of distribution shifts across scientific applications. Specifically, the datasets cover three scientific fields: HEP, biochemistry, and materials science, and are collected from either real experimental scenarios exhibiting distribution shifts, or simulated scenarios designed to mimic real-world distribution shifts. Moreover, we leverage the specific generation process of geometric data, i.e., *the inherent causality* of these applications to categorize their distribution shifts into different categories: conditional shift ($\mathbb{P}_\mathcal{S}(X|Y) \neq \mathbb{P}_\mathcal{T}(X|Y)$ and $\mathbb{P}_\mathcal{S}(Y) = \mathbb{P}_\mathcal{T}(Y)$), covariate shift ($\mathbb{P}_\mathcal{S}(Y|X) = \mathbb{P}_\mathcal{T}(Y|X)$ and $\mathbb{P}_\mathcal{S}(X) \neq \mathbb{P}_\mathcal{T}(X)$), and concept shift ($\mathbb{P}_\mathcal{S}(Y|X) \neq \mathbb{P}_\mathcal{T}(Y|X)$). Furthermore, to address the disparity of assumed available out-of-distribution (OOD) information across previous works, we study three levels: no OOD information (No-Info), only OOD features without labels (O-Feature), and OOD features with a few labels (Par-Label). We evaluate representative methodologies across these three levels, specifically, OOD generalization methods for the No-Info level, domain adaptation (DA) methods for the O-Feature level, and transfer learning (TL) methods for the Par-Label level.

Our experiments are conducted over 6 datasets, in 30 different settings with 10 different distribution shifts times 3 levels of OOD info, covering 3 GDL backbones and 11 learning algorithms in each setting. According to our experiments, we observe that no approach can be the best for all types of shifts, and the levels of OOD info may benefit GDL models to various extents across different applications. In the meantime, our comprehensive evaluation also yields three valuable takeaways to guide the selection of practical solutions depending on the availability of OOD data:

- For Par-Label level, TL strategies show advantages under concept shifts, particularly when there are significant changes in the marginal label distribution.
- For O-Feature level, DA strategies excel when the distribution shifts happen to the geometric characteristics of features that are critical for label determination compared with other features.
- For No-Info level, OOD generalization methods will have some improvements if the training dataset can be properly partitioned into valid groups that reflect the shifts.

In addition to offering domain practitioners guidance on handling distribution shift issues, our new proposed HEP datasets and 10 curated distribution shift scenarios can also facilitate the development and evaluation of new algorithms within the GDL community for various scientific applications.

## 2   Comparison with Existing Benchmarks on Distribution Shifts

Prior research has constructed benchmarks tailored to diverse research fields, shifts, and knowledge levels, and some representative works are summarized in Table 1. In this section, we discuss how GeSS is compared to existing distribution-shift benchmarks in the following three perspectives.

**Application Domain.** Recent works have introduced benchmarks on distribution shifts across various application domains, including tabular data [23, 49], CV [34, 30, 109], OCR [43], GraphML [27, 14, 5], NLP [95, 104], and LLMs [76]. Regarding ML4S, OOD issues have been discussed across various prediction tasks, such as retrosynthesis predictions [102] and property predictions on drugs [35, 86, 108], proteins [41], and materials [56], most of which are built upon GraphML settings. However, no benchmark studies have been conducted in numerous scientific applications, let alone with a focus on GDL models and a broad range of methodologies to deal with distribution shifts like this benchmark.

Table 1: Comparison with existing benchmarks under distribution shifts from three perspectives: Application Domain, Distribution Shift, and Available OOD Info. "Available OOD Info" refers to what type of OOD-Info has been used in the evaluation of these benchmarks.

| Benchmark | Application Domain | Distribution Shift | | | Available OOD Info | | |
|---|---|---|---|---|---|---|---|
| | | Covariate | Conditional | Concept | No-Info | O-Feature | Par-Label |
| WILDS [39], [31] | CV and NLP | ✔ | ✗ | ✗ | ✔ | ✗ | ✗ |
| OoD-Bench [100] | CV | ✔ | ✗ | ✔ | ✔ | ✗ | ✗ |
| WILDS 2.0 [69], [103] | CV and NLP | ✔ | ✗ | ✗ | ✗ | ✔ | ✗ |
| Wild-Time [98] | CV and NLP | ✔ | ✗ | ✔ | ✔ | ✗ | ✔ |
| IGLUE [8] | NLP | ✔ | ✗ | ✗ | ✗ | ✗ | ✔ |
| GOOD [27], GDS [14] | Graph ML | ✔ | ✗ | ✔ | ✔ | ✗ | ✗ |
| DrugOOD [35] | ML4S (Graph ML) only Biochemistry | ✔ | ✗ | ✔ | ✔ | ✗ | ✗ |
| **GeSS (Ours)** | ML4S (GDL) HEP, Biochemistry and Materials Science | ✔ | ✔ | ✔ | ✔ | ✔ | ✔ |

**Distribution Shift**. Previous works explored various distribution shifts. [77, 39, 28, 14] benchmark domain generalization methods. [39, 97, 70] study subpopulation shift. [100] categorizes and quantifies diversity and correlation shifts. [89] jointly analyzes spurious correlation, low-data drift and unseen data shift. [107, 53] benchmark spurious correlations in more diverse and realistic settings. [49] benchmarks a $Y|X$-shift (*aka.*, concept shift in our work) which is shown to be prevalent in tabular data. [27] specifies covariate and concept shifts on the GraphML setting. However, many scientific application scenarios involve distribution shifts with mechanisms that differ from those mentioned above due to their specific data generation processes. Regarding ML4S, previous OOD benchmarks have proposed several data-split strategies to reflect distribution shifts in realistic scientific scenarios, such as molecular sizes and scaffolds [35], protein sequences and structures [41], and chemical reaction conditions [86]. Compared to these works, our benchmark not only collects datasets that can reflect distribution shifts in real-world scientific challenges but also, from an ML perspective, leverages the inherent causality of the specific geometric data generation processes in these applications to categorize their distribution shifts for an in-depth analysis.

**Available OOD Info.** In addition to the level without any OOD data [39, 27], some studies assume the availability of OOD features and benchmark DA methods [69, 24], while others assume the availability of OOD labels to investigate the model transferability [8, 88, 17, 46]. Compared to previous works, we aim to understand the benefits of different levels of OOD data across various distribution shifts, so our benchmark integrates three information levels.

## 3 Benchmark Design

### 3.1 Distribution Shift Categories

Let $\mathcal{X}$ be the input space, $\mathcal{Y}$ be the output space, $h : \mathcal{X} \rightarrow \mathcal{Y}$ be the labeling rule. Under the OOD assumption, we have joint distribution shift, *i.e.*, $\mathbb{P}_{\mathcal{S}}(X, Y) \neq \mathbb{P}_{\mathcal{T}}(X, Y)$. We denote $f(\cdot; \Theta)$ as the GDL model with parameters $\Theta$, and $\ell : \mathcal{Y} \times \mathcal{Y} \rightarrow \mathbb{R}$ as the loss function. Our objective is to find an optimal model $f^*$ with parameters $\Theta^*$, which can be best generalized to target distribution $\mathbb{P}_{\mathcal{T}}$:

$$\Theta^* = \arg \min_{\Theta} \mathbb{E}_{(X,Y) \sim \mathbb{P}_{\mathcal{T}}}[\ell(f(X; \Theta), Y)] \qquad (1)$$

For an in-depth analysis dedicated to scientific applications studied in this work, we consider the following data model [60, 9, 91, 45, 96]. The input variable $X \in \mathcal{X}$ consists of two disjoint parts, namely the causal part $X_c$ and the independent part $X_i$, which satisfies conditional independence with $Y$ given $X_c$, *i.e.*, $X_i \perp\!\!\!\perp Y | X_c$. Next, we categorize various types of distribution shifts.

First, the above data model satisfies $\mathbb{P}(X, Y) = \mathbb{P}(Y|X)\mathbb{P}(X) = \mathbb{P}(Y|X_c)\mathbb{P}(X)$. Thus, we define covariate and concept shifts as follows.

† Covariate Shift holds if $\mathbb{P}_{\mathcal{S}}(Y|X_c) = \mathbb{P}_{\mathcal{T}}(Y|X_c)$, and $\mathbb{P}_{\mathcal{S}}(X) \neq \mathbb{P}_{\mathcal{T}}(X)$.

† Concept Shift holds if $\mathbb{P}_{\mathcal{S}}(Y|X_c) \neq \mathbb{P}_{\mathcal{T}}(Y|X_c)$. Note that the shift of $\mathbb{P}(Y|X_c)$ is also characterized by the change of labeling rule $h$ between $\mathcal{S}$ and $\mathcal{T}$.

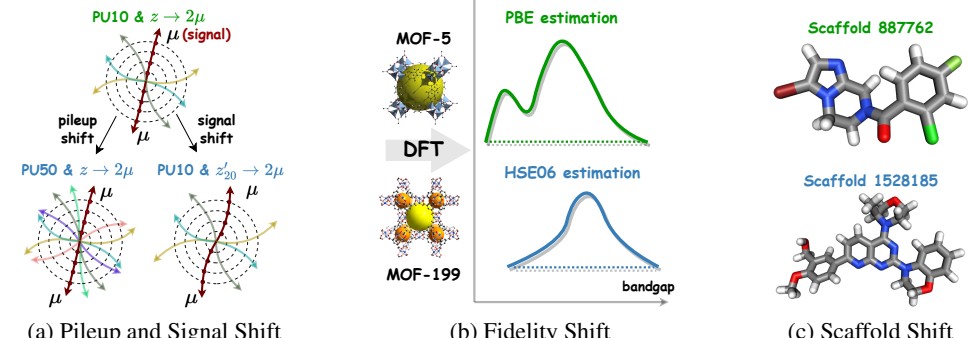

|  (a) Pileup and Signal Shift | (b) Fidelity Shift | (c) Scaffold Shift |

Figure 1: Overview of distribution shifts in this study. The upper (green-colored) and lower (blue-colored) instances represent the scenarios in domains $\mathcal{S}$ and $\mathcal{T}$, respectively. (a) Three-dimensional trajectories of particles in a collision event, which are simulated with a magnetic field parallel to the $z$ axis and plotted on a 2D plane; (b) For the same set of MOFs, the distribution of calculated band gap values exhibits a bimodal (unimodal) nature with lower (higher) expectations under PBE (HSE06) estimation; (c) Molecular three-dimensional stick models with different scaffold IDs across $\mathcal{S}$ and $\mathcal{T}$.

Table 2: Summary of distribution shifts in this study. We also recommend applicable methods for each scenario according to our experimental results, which are shown comprehensively in Table 3.

| Scientific Field | Dataset | Domain | Shift Case | Shift Category | Evaluation Metrics | Applicable Method |
|---|---|---|---|---|---|---|
| HEP | Track-Pileup | Pileup | PU50 PU90 | $\mathcal{I}$-Conditional Shift | ACC | Mixup |
| | Track-Signal | Signal | $\tau \to 3\mu$ $z'_{10} \to 2\mu$ $z'_{20} \to 2\mu$ | $\mathcal{C}$-Conditional Shift | ACC | DANN |
| Materials Science | QMOF | Fidelity | HSE06 HSE06* | Concept Shift | MAE | TL Methods |
| Biochemistry | DrugOOD-3D-Assay | Assay | lbap-core-ic50-assay | Concept Shift | AUC | GroupDRO |
| | DrugOOD-3D-Size | Size | lbap-core-ic50-size | Covariate Shift | | DA or TL Methods |
| | DrugOOD-3D-Scaffold | Scaffold | lbap-core-ic50-scaffold | Covariate Shift | | TL Methods |

On the other hand, we have $\mathbb{P}(X, Y) = \mathbb{P}(X|Y)\mathbb{P}(Y)$. This induces the scenario of Conditional Shift, which holds if $\mathbb{P}_{\mathcal{S}}(X|Y) \neq \mathbb{P}_{\mathcal{T}}(X|Y)$ and $\mathbb{P}_{\mathcal{S}}(Y) = \mathbb{P}_{\mathcal{T}}(Y)$, and Label Shift if $\mathbb{P}_{\mathcal{S}}(X|Y) = \mathbb{P}_{\mathcal{T}}(X|Y)$ and $\mathbb{P}_{\mathcal{S}}(Y) \neq \mathbb{P}_{\mathcal{T}}(Y)$. But as label shift does not arise in our datasets, we later only focus on Conditional Shift. The conditional probability can be decomposed into two parts due to our data model, *i.e.*, $\mathbb{P}(X|Y) = \mathbb{P}(X_c|Y)\mathbb{P}(X_i|X_c)$. This enables us to further categorize Conditional Shift into two sub-types based on the specific factor experiencing shifts, and we observe that these two sub-types exhibit distinct characteristics in our experiments.

† $\mathcal{I}$-Conditional Shift holds if $\mathbb{P}_{\mathcal{S}}(X_i|X_c) \neq \mathbb{P}_{\mathcal{T}}(X_i|X_c)$, and $\mathbb{P}_{\mathcal{S}}(X_c|Y) = \mathbb{P}_{\mathcal{T}}(X_c|Y)$.

† $\mathcal{C}$-Conditional Shift holds if $\mathbb{P}_{\mathcal{S}}(X_i|X_c) = \mathbb{P}_{\mathcal{T}}(X_i|X_c)$, and $\mathbb{P}_{\mathcal{S}}(X_c|Y) \neq \mathbb{P}_{\mathcal{T}}(X_c|Y)$.

Given that each category mentioned above has only one factor experiencing shifts, we naturally partition sub-groups within the source domain $\mathcal{S}$ based on the specific factor undergoing changes.

Note that our considered data models as above do not aim to cover all possible causal relationships. Some other models studied in previous literature [1, 11, 4] are discussed in Appendix A.1, but they do not correspond to GDL applications and thus fall outside the scope of this study. Our data models best describe the mechanisms of applications studied in this work. Besides, the categorization of distribution shifts is determined by the shifted probability term which *directly* manifests through the data generation processes of the studied applications. These processes typically align with scientific experiment procedures or domain-specific theories. More details are in Appendix A.2.

### 3.2 Dataset Curation and Shift Creation

In this section, we introduce the datasets in this study. Table 2 gives a summary. For each dataset, we first introduce the significance of its associated scientific application. Then, we delve into how the distribution shift of each dataset comes from in practice, and categorize the distribution shift according to the definition in Sec. 3.1. Additionally, we elaborate on the selection of domains $\mathcal{S}$ and $\mathcal{T}$, along with partitioning subgroups in the source domain $\mathcal{S}$ for our later experimental setup.

### 3.2.1 `Track`: Particle Tracking Simulation — High Energy Physics

**Motivations.** ML techniques have long been employed and have played a significant role in diverse applications of particle physics [63], including particle flow reconstruction [37], jet tagging [61], and pileup mitigation [54], *etc*. Typically, ML models rely on simulation data for training due to the scarcity of labeled data from real-world experiments. However, the intricate and time-varying nature of experimental environments often leads to distinct physical characteristics that differ from simulated data used for training. For example, the pileup (PU) level, is defined as the number of noisy collisions around the primary collision in Large Hadron Collider experiments [32]. The PU level during the real deployment phase can differ from the PU level used to train the ML model.

**Dataset.** We create `Track`, a particle tracking simulation dataset, and propose `Track-Pileup` and `Track-Signal` datasets. A data sample corresponds to a collision event, which generates numerous particles. Each particle will leave multiple detector hits when traversing the detector. Each point in a data sample represents a detector hit generated by a particle associated with a 3D coordinate. The task is to predict the existence of a specific decay of interest (referred to as *signal* in our work) in a given event, denoted by $Y$, like the decay of $z \to \mu\mu$. This can be formulated as a binary classification task in differentiating the detector hits left by the signal particles plus backgrounds ($X_c + X_i$) from those only left by background particles ($X_i$). This application scenario naturally involves a data generation process $Y \to X$ because the detector hit patterns are determined by whether some type of collision happens. Such generation process leads to conditional shift: The variation in the number of PU particles (Pileup Shift) causes a shift in $\mathbb{P}(X_i|X_c)$, and the change in the type of signal particles (Signal Shift) causes a shift in $\mathbb{P}(X_c|Y)$. We further categorize the two shifts as follows.

**Pileup Shift — $\mathcal{I}$-Conditional Shift.** As shown in the bottom left of Fig. 1a, a higher PU level results in more background particle tracks in a collision while keeping the signal particle track the same. This mechanism aligns with our definition of $\mathcal{I}$-Conditional Shift as $\mathbb{P}_{\mathcal{S}}(X_i|X_c) \neq \mathbb{P}_{\mathcal{T}}(X_i|X_c)$ and $\mathbb{P}_{\mathcal{S}}(X_c|Y) = \mathbb{P}_{\mathcal{T}}(X_c|Y)$. We train the model on source-domain data with the PU level of 10 (`PU10`) and evaluate its generalizability on `PU50` and `PU90` target-domain data, respectively. The division of source-domain subgroups is based on the number of points present in the data entry (one collision event) as it can mimic Pileup Shift in terms of varying particle counts across different PU levels.

**Signal Shift — $\mathcal{C}$-Conditional Shift.** As depicted in the bottom right of Fig. 1a, we alter the geometric characteristics of signal tracks by introducing signal particles with varying momenta, which leads to changes in the *curvature* of signal tracks, while leaving the background particle tracks unchanged. Therefore, we categorize this shift as $\mathcal{C}$-Conditional Shift, as it satisfies $\mathbb{P}_{\mathcal{S}}(X_i|X_c) = \mathbb{P}_{\mathcal{T}}(X_i|X_c)$ and $\mathbb{P}_{\mathcal{S}}(X_c|Y) \neq \mathbb{P}_{\mathcal{T}}(X_c|Y)$. We train the model on source-domain data, where the positive samples consist of 5 different types of signal decays, all characterized by large signal track radii, making them easier to distinguish from background tracks. We evaluate the model on target-domain data with signal decays of $z'_{20} \to 2\mu, z'_{10} \to 2\mu, \tau \to 3\mu$, respectively, whose radii of signal tracks are smaller. We split the source $\mathcal{S}$ into 5 sub-groups, each corresponding to a specific type of signal decay.

### 3.2.2 `QMOF`: Quantum Metal-organic Frameworks — Materials Science

**Motivations.** Materials property prediction plays a crucial role in discovering new materials with favorable properties [62, 93]. Training ML models using data with labels calculated from theory-grounded methods, such as DFT [57], to predict important materials properties, such as band gap, has been an emerging trend, accelerating the process of materials discovery. Among DFT methods, PBE techniques are popular for their cost-effectiveness. However, they are noted for producing low-fidelity results, particularly in the underestimation of band gaps [111, 7]. Conversely, high-fidelity methods exhibit highly accurate calculations but come at the cost of extensive computational resources, resulting in a scarcity of high-fidelity labeled data. Hence, there's a need for methods that allow ML models trained on low-fidelity data to generalize to high-fidelity prediction.

**Dataset.** We introduce the Quantum MOF (`QMOF`) [66, 65], a publicly available dataset comprising over 20,000 metal-organic frameworks (MOFs), coordination polymers, and their quantum-chemical properties calculated from high-throughput periodic DFT. Each point in a sample represents an atom associated with a 3D coordinate. The task is to predict the band gap value of a given material as a regression problem that can be evaluated with MAE metrics. The dataset includes band gap values calculated by 4 different DFT methods (`PBE`, `HLE17`, `HSE06*`, and `HSE06`) ranging from low-fidelity to high-fidelity over the same set of input materials. This naturally forms the shifts across DFT methods at different fidelity levels (Fidelity Shift) categorized as follows.

**Fidelity Shift — Concept Shift.** As illustrated in Fig. 1b, DFT methods at different fidelity levels tend to yield varying distributions of the band gap estimation $Y$ given the same set of input data $X$, thus reflecting the shift of $\mathbb{P}(Y|X)$ characterized by concept shift. Namely, the expected estimation band gap values tend to increase sequentially from PBE (the lowest estimation) to HLE17, HSE06*, and HSE06 (the highest estimation). We construct 2 separate shift cases: one with HSE06 as the target domain $\mathcal{T}$ and the other with HSE06* as the target domain. In both cases, the remaining three levels are used as the source domain $\mathcal{S}$, each serving as a subgroup in the source-domain splits.

### 3.2.3 `DrugOOD-3D`: 3D Conformers of Drug Molecules — Biochemistry

**Motivations.** ML techniques have been applied to various biochemical scenarios, such as protein design [3], molecular docking [12], *etc.*, thereby catalyzing the process of drug discovery. Despite the success, the performance of ML-aided drug discovery easily degrades due to the underlying distribution shifts. Unpredictable public health events like COVID-19 may introduce entirely new targets from unseen domains. Besides, the assay environments, where biochemical properties are measured, may also largely diverge. These challenges related to the distribution shift spur a need for generalizable ML models to further advance drug discovery.

**Dataset.** We adapt DrugOOD [35] and propose `DrugOOD-3D`, with our main focus on the geometric structure of molecules and GDL models. We adopt the task of Ligand Based Affinity Prediction (LBAP) in predicting the binding affinity of a given ligand molecule. We transition the task to a binary classification problem, using AUC scores as evaluation metrics, following DrugOOD. We built `DrugOOD-3D-Scaffold`, `DrugOOD-3D-Size`, and `DrugOOD-3D-Assay` datasets, corresponding to shift cases of `lbap-core-ic50-scaffold`, `lbap-core-ic50-size`, and `lbap-core-ic50-assay`, which cover scaffold, assay, and size shifts introduced as follows.

**Scaffold & Size Shift — Covariate Shift.** The scaffold pattern, illustrated in Fig. 1c, is a significant structural characteristic to describe the core structure of a molecule [101]. Analogously, molecular size is also an important biochemical characteristic. We categorize the two shifts as covariate shifts because the shift in scaffold and size primarily reflects a shift in the marginal input distribution $\mathbb{P}(X)$ across domains, while the labeling rule $h$ and $\mathbb{P}(Y|X)$ are kept invariant.

**Assay Shift — Concept Shift.** We classify assay shift as concept shift since shifts in assays can be viewed as modifying the experimental procedures and conditions. Such modifications could alter the resulting binding affinity value for the same set of molecules, described as a change in $\mathbb{P}(Y|X)$. Note that we follow the same design of domain splits and sub-group splits as DrugOOD.

## 4 Experiments

### 4.1 Experimental Settings

We briefly introduce the experimental settings and leave more details about dataset splits in Appendix C, backbones and learning algorithms in Appendix D, and hyperparameter tuning in Appendix F.

**Backbones.** We include three GDL backbones widely used in various scientific applications: EGNN [71], DGCNN [85], and Point Transformer [110].

**Learning Algorithms.** We select **11** most representative OOD methods to compare. These methods cover general, GNN-grounded, and GDL-grounded algorithms, and span a broad range of learning strategies under different levels of OOD info, *i.e.*, No-Info, O-Feature, and Par-Label levels. For **No-Info** level, we select 1) *vanilla*: ERM [82]; 2) *invariant learning*: VREx [40]; 3) *data augmentation*: MixUp [105]; 4) *subgroup robust method*: GroupDRO [68]; 5) *causal inference*: DIR [91]; 6) *information bottleneck*: LRI [55]. Note that DIR is a well-known graph-based OOD baseline and LRI is a novel algorithm grounded in GDL. We refer to the above-mentioned methods as *OOD generalization methods* for simplicity. For **O-Feature** level, we select *domain-invariant methods*: 7) DANN [22] and 8) DeepCoral [75]. For **Par-Label** level, we conduct *full* fine-tuning, *i.e.*, all model parameters get fine-tuned, with 9) 100, 10) 500, and 11) 1000 labels, denoted as $TL_{100}$, $TL_{500}$, $TL_{1000}$ respectively. Regarding Fidelity Shift, we select a subset of OOD generalization methods (VREx and GroupDRO) that are compatible with regression tasks to evaluate.

We provide a detailed discussion on the rationale behind the selection of methods and GDL backbones in Appendices D.1 ad D.2. Additionally, we have developed a highly modular codebase, allowing for the seamless evaluation of new algorithms tailored for the GDL setting using this benchmark.

Table 3: Experimental results (Test-ID and Test-OOD performance) on **Pileup** (PU50 and PU90 cases), **Signal** ($\tau \to 3\mu$ and $z'_{10} \to 2\mu$ cases), **Size**, **Scaffold**, and **Fidelity** (HSE06 and HSE06* cases) shifts over EGNN and DGCNN. Note that Test-ID performance of TL methods is not evaluated. Parentheses show standard deviation across 3 replicates. ↑ denotes higher values correspond to better performance, whereas ↓ denotes lower for better. We **bold** and underline the best and the second-best OOD performance, and use † to mark best within the No-Info level for each distribution shift scenario.

### Pileup Shift — $\mathcal{I}$-Conditional Shift (ACC↑)

| | | EGNN | | | | DGCNN | | | |
|---|---|---|---|---|---|---|---|---|---|
| | | PU50 | | PU90 | | PU50 | | PU90 | |
| Level | Algorithm | Test-ID | Test-OOD | Test-ID | Test-OOD | Test-ID | Test-OOD | Test-ID | Test-OOD |
| No-Info | ERM | 95.75(0.08) | 87.65(0.30) | 96.11(0.15) | 80.99(1.40) | 94.35(0.42) | 86.56(0.96)† | 94.35(0.42) | 79.84(1.08) |
| | VREx | 95.49(0.32) | 87.45(0.76) | 95.49(0.32) | 80.77(0.93) | 94.54(0.17) | 86.37(0.84) | 94.54(0.17) | 80.41(0.91)† |
| | GroupDRO | 93.18(0.33) | 83.03(0.32) | 93.18(0.33) | 75.67(0.63) | 91.48(0.19) | 79.38(0.59) | 91.48(0.19) | 73.69(0.37) |
| | DIR | 95.10(0.09) | 85.59(0.45) | 95.10(0.09) | 78.47(0.17) | 94.01(0.29) | 84.33(0.65) | 94.01(0.29) | 75.73(1.39) |
| | LRI | 95.80(0.14) | 88.15(0.31) | 95.77(0.43) | 81.43(0.80) | 93.95(0.04) | 85.25(0.04) | 93.95(0.04) | 78.65(0.39) |
| | MixUp | 95.78(0.41) | 89.41(0.11)† | 95.86(0.13) | 82.29(0.40)† | 94.18(0.25) | 85.93(0.46) | 94.18(0.25) | 79.43(0.77) |
| O-Feature | DANN | 95.18(0.51) | 87.16(0.72) | 95.86(0.10) | 80.69(0.44) | 93.91(0.47) | 85.01(0.64) | 94.33(0.12) | 76.15(1.95) |
| | Coral | 95.13(0.27) | 86.98(0.80) | 95.19(0.07) | 78.99(1.79) | 94.17(0.21) | 84.61(1.04) | 94.66(0.16) | 77.08(0.94) |
| Par-Label | $TL_{100}$ | | 84.20(0.46) | | 77.85(0.59) | | 81.65(1.06) | | 73.48(0.39) |
| | $TL_{500}$ | | 87.05(0.46) | | 82.09(0.88) | | 84.41(1.06) | | 78.19(0.94) |
| | $TL_{1000}$ | | 87.61(0.13) | | **83.40(0.80)** | | 85.09(0.70) | | 79.97(0.38) |

### Signal Shift — $\mathcal{C}$-Conditional Shift (ACC↑)

| | | EGNN | | | | DGCNN | | | |
|---|---|---|---|---|---|---|---|---|---|
| | | $\tau \to 3\mu$ | | $z'_{10} \to 2\mu$ | | $\tau \to 3\mu$ | | $z'_{10} \to 2\mu$ | |
| Level | Algorithm | Test-ID | Test-OOD | Test-ID | Test-OOD | Test-ID | Test-OOD | Test-ID | Test-OOD |
| No-Info | ERM | 97.15(0.20) | 65.98(0.77) | 96.85(0.23) | 70.72(1.28) | 95.72(0.23) | 65.30(1.03) | 95.37(0.12) | 69.04(0.31) |
| | VREx | 96.86(0.29) | 66.38(0.80) | 96.66(0.15) | 71.46(0.87) | 95.66(0.03) | 64.91(0.47) | 95.49(0.32) | 69.83(0.06) |
| | GroupDRO | 96.48(0.22) | 67.15(0.10) | 96.71(0.08) | 72.56(0.88)† | 95.02(0.06) | 66.01(0.33) | 95.06(0.32) | 69.76(0.03) |
| | DIR | 77.91(2.87) | 67.32(0.43) | 93.56(2.62) | 70.04(1.00) | 91.87(0.53) | 64.74(0.82) | 91.87(0.53) | 70.67(0.81)† |
| | LRI | 96.25(0.16) | 67.49(0.24)† | 96.37(0.21) | 69.91(0.89) | 90.50(0.89) | 67.82(0.06)† | 93.40(0.28) | 67.84(0.07) |
| | MixUp | 96.95(0.22) | 65.63(0.77) | 96.97(0.06) | 71.39(1.49) | 95.41(0.33) | 66.02(1.01) | 95.80(0.08) | 69.23(0.01) |
| O-Feature | DANN | 81.37(1.04) | **68.05(0.09)** | 90.08(0.86) | **77.36(0.83)** | 80.72(1.34) | **68.27(0.36)** | 87.90(0.22) | **75.46(0.53)** |
| | Coral | 96.32(0.67) | 66.61(0.49) | 96.82(0.18) | 71.60(0.42) | 94.93(0.16) | 65.06(0.43) | 94.29(0.04) | 68.95(0.73) |
| Par-Label | $TL_{100}$ | | 64.08(1.01) | | 74.21(0.94) | | 64.37(0.29) | | 63.19(0.03) |
| | $TL_{500}$ | | 67.08(0.04) | | 75.81(0.86) | | 65.99(0.76) | | 66.42(0.45) |
| | $TL_{1000}$ | | 67.47(0.11) | | 77.02(0.37) | | 65.73(0.78) | | 66.03(0.33) |

### Size & Scaffold Shift — Covariate Shift (AUC↑)

| | | EGNN | | | | DGCNN | | | |
|---|---|---|---|---|---|---|---|---|---|
| | | Size | | Scaffold | | Size | | Scaffold | |
| Level | Algorithm | Test-ID | Test-OOD | Test-ID | Test-OOD | Test-ID | Test-OOD | Test-ID | Test-OOD |
| No-Info | ERM | 91.06(0.26) | 64.98(0.54) | 84.73(0.48) | 68.16(0.82) | 89.60(0.04) | 62.56(0.61) | 81.89(0.14) | 67.05(0.49) |
| | VREx | 91.20(0.08) | 65.01(0.50)† | 84.76(0.54) | 68.20(0.31) | 89.41(0.21) | 62.91(0.47) | 82.95(0.43) | 68.24(0.25) |
| | GroupDRO | 86.80(0.23) | 61.11(0.31) | 85.38(0.16) | 68.07(0.66) | 83.41(0.56) | 60.55(0.02) | 83.27(0.25) | 67.57(0.18) |
| | DIR | 87.24(0.84) | 64.40(0.42) | 80.59(2.34) | 67.70(1.19) | 80.43(0.50) | 62.05(0.65) | 74.49(0.37) | 67.19(0.91) |
| | LRI | 91.00(0.32) | 64.05(0.26) | 85.00(0.79) | 67.61(0.26) | 89.50(0.30) | 63.00(0.41) | 80.20(0.75) | 67.69(0.28) |
| | MixUp | 91.02(0.46) | 63.87(0.24) | 85.36(0.29) | 68.28(0.19)† | 89.45(0.19) | 63.65(0.21)† | 82.71(0.57) | 68.33(0.69)† |
| O-Feature | DANN | 91.25(0.05) | **65.45(0.45)** | 85.65(0.42) | 67.66(0.90) | 89.08(0.37) | 63.73(0.49) | 82.30(0.69) | 67.74(0.50) |
| | Coral | 91.32(0.19) | 64.77(0.49) | 85.41(0.72) | 68.61(0.48) | 89.05(0.36) | 63.87(0.48) | 81.92(0.69) | 67.26(0.62) |
| Par-Label | $TL_{100}$ | | 64.48(0.29) | | 67.21(0.34) | | 62.50(0.21) | | 67.10(0.18) |
| | $TL_{500}$ | | 64.84(0.28) | | 68.94(0.67) | | 62.54(0.22) | | 68.15(0.13) |
| | $TL_{1000}$ | | **65.43(0.27)** | | **70.71(0.43)** | | 63.00(0.38) | | **68.79(0.11)** |

### Fidelity Shift — Concept Shift (MAE ↓)

| | | EGNN | | | | DGCNN | | | |
|---|---|---|---|---|---|---|---|---|---|
| | | HSE06 | | HSE06* | | HSE06 | | HSE06* | |
| Level | Algorithm | Test-ID | Test-OOD | Test-ID | Test-OOD | Test-ID | Test-OOD | Test-ID | Test-OOD |
| No-Info | ERM | 0.508(0.003) | 1.099(0.095) | 0.624(0.014) | 0.556(0.007) | 0.486(0.005) | 1.082(0.030) | 0.604(0.003) | 0.547(0.007) |
| | VREx | 0.511(0.005) | 1.083(0.063) | 0.628(0.010) | **0.534(0.012)**† | 0.511(0.002) | 1.042(0.075) | 0.620(0.002) | 0.522(0.007) |
| | GroupDRO | 0.533(0.003) | 0.996(0.029)† | 0.689(0.009) | 0.546(0.002) | 0.515(0.001) | 0.977(0.021)† | 0.698(0.004) | 0.518(0.006)† |
| O-Feature | DANN | 0.502(0.004) | 1.161(0.017) | 0.623(0.011) | 0.570(0.012) | 0.484(0.001) | 1.051(0.030) | 0.603(0.007) | 0.540(0.009) |
| | Coral | 0.504(0.004) | 1.161(0.045) | 0.623(0.005) | 0.571(0.009) | 0.488(0.003) | 1.062(0.014) | 0.605(0.007) | 0.538(0.007) |
| Par-Label | $TL_{100}$ | | 0.732(0.009) | | 0.629(0.036) | | 0.695(0.026) | | 0.603(0.009) |
| | $TL_{500}$ | | 0.638(0.008) | | 0.556(0.013) | | 0.620(0.010) | | 0.541(0.005) |
| | $TL_{1000}$ | | **0.625(0.003)** | | 0.547(0.010) | | **0.575(0.008)** | | **0.517(0.000)** |

**Dataset Splits.** For each dataset, we first divide it into the ID dataset and the OOD dataset based on our characterization of $\mathbb{P}_\mathcal{S}$ and $\mathbb{P}_\mathcal{T}$. The resulting dataset in the source domain contains multiple subgroups following our split covered in Sec. 3.2, for the operation of OOD methods that rely on subgroup splits. Subsequently, the ID and OOD datasets are randomly segmented into Train-ID, Val-ID, and Test-ID, and Train-OOD, Val-OOD, and Test-OOD, respectively.

**Model Training & Evaluation.** For fair comparisons across the three info levels, we meticulously set up both the model training and evaluation processes: In No-Info level, we train the model solely on the Train-ID dataset via OOD methods. In O-Feature level, we apply DA algorithms and train

the model on the *same* Train-ID used in No-Info level ***plus*** the extra data feature info of the entire OOD dataset. In Par-Label level, we use the Train-OOD dataset to fine-tune the model which has already been pre-trained on the *same* Train-ID used in the two previous levels. Although the training datasets may vary due to the need to contain different levels of OOD info, we achieve a fair comparison by keeping Train-ID data invariant. Across all levels of OOD info, we evaluate the model's ID performance using the *same* Val-ID and Test-ID datasets, and its OOD performance using the *same* Val-OOD and Test-OOD datasets. For hyperparameter tuning, we tune a predefined set of hyperparameters and select the model with the best metric score of Val-OOD for the ultimate evaluation. Also, we thoroughly discuss the motivation and insights of our design of model training and evaluation processes, and put details in Appendix E.

## 4.2  Results Analysis — General Tendency

We put experimental results on 2 of 3 backbones in Table 3. Complete results can be found in Appendix G. We begin by presenting overall comparisons and general findings: Although $TL_{1000}$ outperforms ERM by a notable margin in several cases, fine-tuning can sometimes result in negative effects when the labeled OOD data is quite limited, particularly in cases involving a smaller degree of distribution shifts (*cf.* Fig. 2). This is consistent with [38], where fine-tuning a large model based on a small set of labels may lead to catastrophic forgetting. To mitigate this issue, robust fine-tuning strategies, such as weight-space ensembles [90], regularization [94] and surgical fine-tuning [44], could be potential solutions.

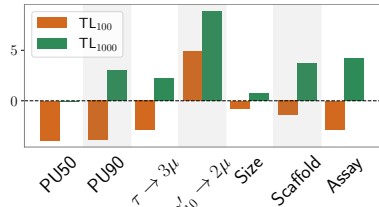

Figure 2: Test-OOD improvements (%) over ERM for $TL_{100}$ and $TL_{1000}$ across different shift cases (in the EGNN backbone).

## 4.3  Results Analysis — Insightful Conclusions

Besides the general observations exhibited above, our experiments also yield some intriguing conclusions that may be widely applicable. We structure this subsection by first presenting our conclusions, exemplified by representative observations and rational explanations.

● **Conclusion 1. DA strategies excel in $\mathcal{C}$-Conditional Shift, where some variation arises in the geometric characteristics of the causal component $X_c$.**

A great example illustrating this conclusion comes from Signal Shift. As shown in Fig. 4, DANN, a DA method, performs particularly well in the case of $z'_{10} \to 2\mu$, largely outperforming ERM (↑ 9.4% in the EGNN model) and all OOD generalization methods without OOD info (↑ at least 6.6% in EGNN). As introduced before, Signal Shift represents a $\mathcal{C}$-conditional shift, which in this case arises from the shift in the curvature of the signal tracks, whose points collectively form the causal component $X_c$. The access to OOD features enables the model to align the latent representations of the causal components with varying distributions of geometries (curvatures here) across the source and target domains, thereby aiding in the correct identification of unseen signal types.

In contrast, Fig. 4 shows that DA strategies yield performance very close to ERM in Pileup Shift. Although both Pileup and Signal shifts are categorized as Conditional Shifts (*cf.* Sec. 3.2.1), they mainly differ in two aspects: 1) Pileup Shift represents an $\mathcal{I}$-Conditional Shift, occurring exclusively in the independent part, *i.e.* $\mathbb{P}(X_i|X_c)$, and 2) it involves a variation in the number of particle tracks rather than geometric characteristics, which is different from Signal Shift. We propose two plausible explanations for the challenges faced by DA strategies in Pileup Shift, centered around these disparities, and recommend interested readers see **H2** in Appendix H.

● **Conclusion 2. TL methods excel under Concept Shift, particularly when the shift of the marginal label distribution $\mathbb{P}(Y)$ is large.**

Here we examine two cases of Fidelity Shift, where the TL strategy demonstrates contrasting results (*cf.* Fig. 3a): TL performs particularly well in the case of HSE06, where it largely outperforms all other methods with the MAE score increased by at least 40%. However, it exhibits only limited improvement in the case of HSE06*. We explain the difference by analyzing the marginal label distributions $\mathbb{P}_{\mathcal{S}}(Y)$ and $\mathbb{P}_{\mathcal{T}}(Y)$. As mentioned before, Fidelity Shift can be characterized by a change in the labeling rule, *i.e.*, two similar inputs may be mapped to very different $Y$ values.

Specifically, fidelity levels of PBE, HLE17, and HSE06* provide estimations that are closer to each other, while the fidelity level of HSE06 significantly exceeds the other three. Therefore, the case with HSE06 as the target domain $\mathcal{T}$ but the other three as the source $\mathcal{S}$, yields a

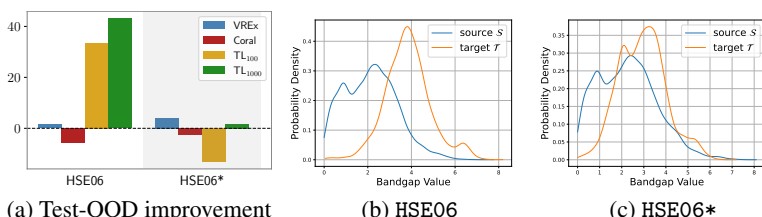

(a) Test-OOD improvement (b) HSE06 (c) HSE06*

Figure 3: (a) Test-OOD improvements (%) over ERM for VREx, DeepCoral, $\mathrm{TL}_{100}$ and $\mathrm{TL}_{1000}$ methods in Fidelity Shifts (including HSE06 and HSE06* cases) in the EGNN backbone; (b)/(c) KDE [67, 58] curves of the marginal label distribution $\mathbb{P}(Y)$ across the source $\mathcal{S}$ and target $\mathcal{T}$ in the cases of HSE06 / HSE06*.

large gap in the distribution of $\mathbb{P}(Y)$ between the two domains, *i.e.,* $\mathbb{P}_{\mathcal{S}}(Y) \neq \mathbb{P}_{\mathcal{T}}(Y)$ (*cf.* Fig. 3b). Therefore, the OOD labels are crucial to finetune the model predictions to match the aimed distribution $\mathbb{P}_{\mathcal{T}}(Y)$. In contrast, the case when HSE06* is the target domain $\mathcal{T}$ but the others as $\mathcal{S}$, yields closer distributions of $\mathbb{P}(Y)$ between the two domains, *i.e.*, $\mathbb{P}_{\mathcal{S}}(Y) \approx \mathbb{P}_{\mathcal{T}}(Y)$ (*cf.* Fig. 3c). Therefore, only using a few OOD labels to finetune the model tends to have a limited impact on its performance.

• **Conclusion 3. For the OOD generalization methods to learn robust representations, the more informatively the groups obtained by splitting the source domain $\mathcal{S}$ indicate the distribution shift, the better performance these methods may achieve.**

This observation is related to GroupDRO, an OOD method that is to learn robust representation across different group splits of the source $\mathcal{S}$. As illustrated in Fig. 4, GroupDRO almost consistently outperforms ERM in all cases with Signal Shift ($\tau$, $z'_{10}$, $z'_{20}$) while it largely under-performs ERM in the cases of Pileup Shift (PU50, PU90). GroupDRO captures robustness by increasing the importance of subgroups with larger errors and thus highly relies on the assumption that the shift between the splits of in-domain data can to some extent reflect the distribution shift between the source $\mathcal{S}$ and the target $\mathcal{T}$. In the cases with Signal Shift, the way to split subgroups of the source domain aligns well with the distribution shift: Each split represents a distinct type of decay (5 types in total). By learning robust representations across these subgroups, GroupDRO yields better OOD generalization. In contrast, in the case of

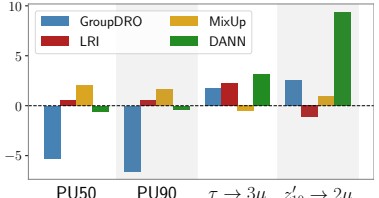

Figure 4: Test-OOD improvements (%) over ERM for GroupDRO, MixUp, LRI, and DANN methods across Pileup Shifts (cases of PU50/90) and Signal Shifts (cases of $\tau \to 3\mu$ and $z'_{10} \to 2\mu$) in the EGNN backbone.

Pileup Shift, varying the number of points in a collision event is used as a proxy of the shift to achieve the group splits of the training dataset, based on the fact that the PU level is positively correlated with the number of particles. This way of subgroup splits is subjective, which is limited by the availability of data and may not fully reflect Pileup Shift between domain $\mathcal{S}$ and $\mathcal{T}$.

More analysis (obervations, conclusions, and conjectures) of experimental results are put in Appendix H. Also, we identify conclusions that align with existing OOD literature and others that are novel and *specific* to the GDL setting. We conduct comparisons (including consistency and disparity) between our findings and previous ones in Appendix I.

## 5 Conclusion

This work systematically evaluates the performance of GDL models when scientific applications meet distribution shifts. Our benchmark has 30 distinct scenarios with 10 shift cases times 3 levels of available OOD info, covering 3 GDL backbones and 11 learning algorithms. Based on our evaluation, we reveal several intriguing discoveries. In particular, our results may help select applicable solutions based on the causal mechanism behind the distribution shift and the availability of OOD info. Moreover, our work encourages more realistic and rigorous evaluations of GDL used in scientific applications, and may inspire methodological advancements for GDL to deal with distribution shifts.

## Acknowledgments

The authors would like to thank Dr. Callie Hao and the Sharc Lab at Georgia Tech. for their computing resources (NVIDIA RTX A6000) to support this work, and also thank Dr. Mia Liu at Purdue University for suggestions about distribution shifts in HEP, and Dr. Jan-Frederik Schulte at Purdue University for the help with the generation of some of HEP data. Deyu Zou, Shikun Liu, Siqi Miao and Pan Li are partially supported by the NSF awards PHY-2117997, IIS-2239565, IIS-2428777, and CCF-2402816; DOE award DE-FOA-0002785; JPMC faculty awards; OpenAI Researcher Access Program Credits; and Microsoft Azure Research Credits for Generative AI.

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

# A More Discussions about Distribution Shift Formulations

## A.1 Details of the Data Model

Here we give a complementary discussion about the established data model that are discussed in Sec. 3.1. Concretely, we adopt the Structure Causal Model (SCM) [59, 60] to represent $\mathcal{I}$-conditional, $\mathcal{C}$-conditional, covariate, and concept shifts from a causal view. As illustrated in Fig. 5, five variables, including input $X$, causal part $X_c$, independent part $X_i$, domain $D$ and label $Y$, are linked by the direct causal correlation "$\rightarrow$". For some variable $A$ in the SCM, $\mathbf{Pa}(A) \rightarrow A$ denotes as the direct causal link from its parent variables $\mathbf{Pa}(A)$ to $A$. According to the causal theory [59, 60], there exists the correlation $\mathbf{Pa}(A) \rightarrow A$, if and only if there exists a function $f_A$, *s.t.*, $A = f_A(\mathbf{Pa}(A), \epsilon_A)$, where $\epsilon_A$ is exogenous noise satisfying $\epsilon_A \perp\!\!\!\perp \mathbf{Pa}(A)$, and we omit the exogenous noise in this study for simplification. Plus note we treat $D$ as an additional variable that exerts an influence on the other variables and thus induces a shift in the corresponding probability distribution between the domains $\mathcal{S}$ and $\mathcal{T}$.

We start with the correlation that is shared across four categories. The input variable $X$ consists of two disjoint parts $X_i$ and $X_c$, *i.e.*, $X_i \rightarrow X \leftarrow X_c$. For a convenient analysis of the proposed application scenarios, we introduce certain assumptions or specifications. However, it's important to note that our data model and its associated assumptions do not aim to cover all possible causal relationships. To provide a more comprehensive view, each specification or assumption is accompanied by some examples (from other works) that challenge it.

- We do not further discuss potential causal dependencies between $X_i$ and $X_c$ for simplicity and use a dashed arrow to represent such potential dependencies, following [91]. However, some works [1, 11] involved the explicit discussions of latent interactions between $X_i$ and $X_c$.
- We assume the independent part $X_i$ and the label variable $Y$ to be conditionally independent given $X_c$, *i.e.*, $X_i \perp\!\!\!\perp Y|X_c$, as discussed in Sec. 3.1. Also, there exists some causal relationships which violate this assumption of conditional independence. A typical example lies in the PIIF SCM[4, 1], where the correlation $X_c \rightarrow Y \rightarrow X_i$ breaks the assumption. However, we haven't identified a scenario in this work that adheres to such a setting.

In particular, the causal part $X_c$ shares the causal correlation with $Y$, represented as either $X_c \rightarrow Y$ (which is assumed by many previous works), or $Y \rightarrow X_c$ (which appears in our study), corresponding to the aforementioned data generating process $X \rightarrow Y$ and $Y \rightarrow X$.

## A.2 Details of the Shift Categories

Concretely, we classify the following distribution shifts based on their distinct data generation process between $X$ and $Y$ (specifically, the correlation between $X_c$ and $Y$) as well as how the domain variable $D$ affects individual variables like $X_i, X_c$ or $Y$. Note that we follow the well-established definitions of these shifts and further extend the definitions to what we present in our work based on our established data model to better align with the application scenarios we propose.

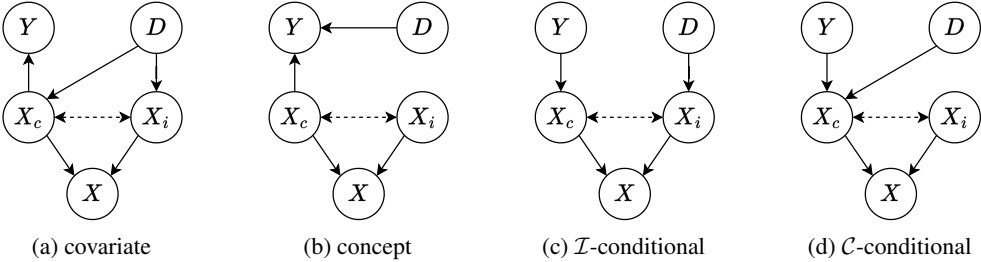

Figure 5: SCMs of covariate, concept, $\mathcal{I}$-conditional, and $\mathcal{C}$-conditional shifts.

**Covariate Shift** [26] is initially defined as $\mathbb{P}_{\mathcal{S}}(X) \neq \mathbb{P}_{\mathcal{T}}(X)$ and $\mathbb{P}_{\mathcal{S}}(Y|X) = \mathbb{P}_{\mathcal{T}}(Y|X)$. In the context of our work, the data generating process of $X \rightarrow Y$ induces the assumption of covariate and concept shifts, and covariate shift holds if $\mathbb{P}_{\mathcal{S}}(X) \neq \mathbb{P}_{\mathcal{T}}(X)$ and $\mathbb{P}_{\mathcal{S}}(Y|X_c) = \mathbb{P}_{\mathcal{T}}(Y|X_c)$, which are

achieved by the variable $D$ exclusively impacting $X$ without affecting $Y$, as shown in Fig. 5a. Note that we do not further specify which part of $X$, $X_c$ or $X_i$, is impacted by the variable $D$. This is in line with real-world scenarios where the specific shifts in $\mathbb{P}(X_c)$ or $\mathbb{P}(X_i)$ may not be apparent, such as in our instances of the scaffold shift and size shift based on the DrugOOD-3D dataset. Therefore, we present both $D \to X_c$ and $D \to X_i$ in Fig. 5a for simplicity.

**Concept Shift** [52, 21] is initially formalized as $\mathbb{P}_{\mathcal{S}}(Y|X) \neq \mathbb{P}_{\mathcal{T}}(Y|X)$. Under our formulation, concept shift holds if $\mathbb{P}_{\mathcal{S}}(Y|X_c) \neq \mathbb{P}_{\mathcal{T}}(Y|X_c)$. This is characterized by the correlation $D \to Y \leftarrow X_c$, as shown in Fig. 5b, which means the label $Y$ is determined collectively by the input causal part $X_c$ and the domain variable $D$, and more importantly, there is a change of the labeling rule $h$ across domains $\mathcal{S}$ and $\mathcal{T}$. Note that in our study we do not further assume if the shift in $\mathbb{P}(X)$ exists. In the fidelity shift, DFT methods with varying fidelity levels calculate band gap values of the same set of MOFs, where we consider that $\mathbb{P}(X)$ remains invariant across domains, while the assay shift, also categorized as concept shift, may involve the shift in $\mathbb{P}(X)$. So we do not explicitly present the correlation between $D$ and $X_i$ or $X_c$ in Fig. 5b.

**Conditional Shift**, as proposed in [106], is induced by the data generating process of $Y \to X$ and holds if $\mathbb{P}_{\mathcal{S}}(X|Y) \neq \mathbb{P}_{\mathcal{T}}(X|Y)$. We follow this formulation in our work. Note that $Y \to X$ aligns well with the scenario of the Track dataset, where the simulated physical event $X$ is controlled by multiple parameters, including one representing the label $Y$ as positive or negative. The conditional distribution could be decomposed into two distinct parts based on the data model of $X_i \perp\!\!\!\perp Y|X_c$: $\mathbb{P}(X|Y) = \mathbb{P}(X_c|Y)\mathbb{P}(X_i|X_c, Y) = \mathbb{P}(X_c|Y)\mathbb{P}(X_i|X_c)$, which serves as the basis to further categorize conditional shift into the two following sub-types.

- $\mathcal{I}$-**Conditional Shift** holds if $\mathbb{P}_{\mathcal{S}}(X_i|X_c) \neq \mathbb{P}_{\mathcal{T}}(X_i|X_c)$ and $\mathbb{P}_{\mathcal{S}}(X_c|Y) = \mathbb{P}_{\mathcal{T}}(X_c|Y)$. As shown in Fig. 5c, the domain variable $D$ exclusively affects the independent part $X_i$, $i.e.$, $D \to X_i$. In this case, only the conditional distribution $\mathbb{P}(X_i|X_c)$ changes across domains $\mathcal{S}$ and $\mathcal{T}$. Note that there does not exist the causal link of $X_i \to X_c$ in this scenario to hold the assumption of $X_i \perp\!\!\!\perp Y|X_c$, so the distribution $\mathbb{P}(X_c|Y)$ will not be indirectly influenced by $D$ and thus keeps invariant across the domains.

- $\mathcal{C}$-**Conditional Shift** holds if $\mathbb{P}_{\mathcal{S}}(X_i|X_c) = \mathbb{P}_{\mathcal{T}}(X_i|X_c)$ and $\mathbb{P}_{\mathcal{S}}(X_c|Y) \neq \mathbb{P}_{\mathcal{T}}(X_c|Y)$. As shown in Fig. 5d, the domain variable $D$ exclusively affects the causal part $X_c$, which forms the structure of $Y \to X_c \leftarrow D$, representing the distribution of $X_c$ is determined by both $Y$ and $D$. That means only the conditional distribution $\mathbb{P}(X_c|Y)$ changes across the domains $\mathcal{S}$ and $\mathcal{T}$ while the distribution $\mathbb{P}(X_i|X_c)$ keeps invariant.

**Discussions**. In terms of covariate shift and concept shift, our extended formulations, compared to the initial definitions, put greater emphasis on the analysis of $\mathbb{P}(Y|X_c)$ rather than $\mathbb{P}(Y|X)$, due to the assumption of conditional independence in our data model. Such extension is beneficial because the underlying rationale or causal correlation, often rooted in well-established scientific rules or theories like Density Functional Theory (DFT), holds significant importance for scientific discovery and ML4S. Therefore, it deserves more attention.

### A.3 Additional Comparisons with Distribution Shifts in Related Studies

Here we conduct additional comparisons with some distribution shifts which have been proposed by related works.

**Concept Shift.** [27] and [100] also involved shift cases explicitly formalized as concept shift. However, we claim it operates under a different mechanism than the one formalized in our study. In our work, concept shift particularly denotes the change in causal correlation between $X_c$ and $Y$, $i.e.$, the shift in $\mathbb{P}(Y|X_c)$. This definition aligns perfectly with a real-world scenario of fidelity shift observed in materials science. In contrast, in GOOD's context [27], concept shift corresponds to changes in statistical rather than causal correlation. For instance, it may involve correlations between color and digit in datasets like ColoredMNIST [27, 14].

**Covariate Shift.** Diversity shift [100], low-data drift [89], and unseen data shift [89] can be explicitly formalized as covariate shift. However, our work does not aim to conduct a more find-grained analysis within a single shift category. Instead, our goal is to formalize and categorize diverse distribution-shift mechanisms across various scientific application scenarios.

Table 4: Dataset statistics for each dataset and distribution shift scenario. We evaluate the ID performance of models using Val-ID and Test-ID Datasets, and the OOD performance using Val-OOD and Test-OOD Datasets. The "OOD" column in this table presents the total number of OOD data entries whose features (but not labels) are used in the O-Feature level. Note that this table does not include statistics of Train-OOD that is specifically used for fine-tuning models in the Par-label level as mentioned in Sec. 4.1, because we utilize a fixed number of 100, 500, 1000 labels in this case, corresponding to $\text{TL}_{100}$, $\text{TL}_{500}$, $\text{TL}_{1000}$ baselines respectively. For bi-classification tasks, we list the number of positive data points (left) and negative data points (right), separated by "/".

| Dataset | Shift | Shift Case | Train-ID | Val-ID | Test-ID | OOD | Val-OOD | Test-OOD |
|---|---|---|---|---|---|---|---|---|
| Track-Pileup | Pileup | PU50 | 14814/15634 | 2469/2605 | 2470/2607 | 10000/10000 | 2500/2500 | 2500/2500 |
| | | PU90 | | | | 7700/7700 | 2500/2500 | 2500/2500 |
| Track-Signal | Signal | $\tau \to 3\mu$ | 11851/15000 | 1975/2500 | 1975/2500 | 12000/15000 | 1975/2500 | 1977/2500 |
| | | $z'_{10} \to 2\mu$ | | | | 12000/15000 | 1975/2500 | 1977/2500 |
| | | $z'_{20} \to 2\mu$ | | | | 12000/15000 | 1975/2500 | 1977/2500 |
| QMOF | Fidelity | HSE06 | 10781 | 1796 | 1798 | 6000 | 2000 | 2000 |
| | | HSE06* | 10781 | 1796 | 1798 | 6000 | 2000 | 2000 |
| DrugOOD-3D-Assay | Assay | lbap-core-ic50-assay | 29060/3861 | 9611/1295 | 9945/1323 | 32371/4687 | 17099/1557 | 15272/3130 |
| DrugOOD-3D-Size | Size | lbap-core-ic50-size | 32686/2542 | 10872/857 | 11003/846 | 26426/6921 | 14657/2706 | 11769/4215 |
| DrugOOD-3D-Scaffold | Scaffold | lbap-core-ic50-scaffold | 19455/1116 | 4473/211 | 26670/3015 | 30389/6824 | 16020/2678 | 14369/4146 |

# B  Preliminaries for Geometric Deep Learning

**Notations**. We consider a geometric data sample $g = (\mathcal{V}, \mathbf{X}, \mathbf{r})$, where $\mathcal{V} = \{v_1, \cdots, v_n\}$ is a set of points with the size $n$, $\mathbf{X} \in \mathbb{R}^{n \times m}$ denotes as $m$-dimensional point features, and $\mathbf{r} \in \mathbb{R}^{n \times d}$ denotes as $d$-dimensional spacial coordinates of points. We specifically focus on 3D coordinates of scientific data in our study, *i.e.*, $d = 3$. We build the GDL model $\hat{y} = f(g; \Theta)$ to predict the ground-truth label $y$ of data $g$, where $y$ is categorical for classification tasks and continuous for regression tasks. The model in our study consists of two parts, *i.e.*, $f = \omega \circ \Phi$, including the GDL component $\Phi$, which is based on multiple GDL layers, and the MLP component $\omega$, which gives the final prediction. And we hope the GDL models maintain strong predictive performance even when $g$ is drawn from a distribution differing from the one during training, which motivates our study.

**Pipelines**. Here we present *how GDL backbones handle geometric data* in this study. Given $N$ samples of $\{g_i\}_{i=1}^N$, we begin with constructing a $k$-nn graph for each data entry based on the spacial distances, *i.e.*, $\|\mathbf{r}_v - \mathbf{r}_u\|_2$ between any pair of points $u, v \in \mathcal{V}$, where $k$ is a hyperparameter. The GDL model then iteratively updates the representation of the point $v$ via aggregation $\text{AGG}_{u \in \mathcal{N}(v)}(\mathbf{m}_{uv})$, where AGG denotes as the aggregation operator (*e.g.*, $\sum$ or max), $\mathcal{N}(v)$ denotes as the neighbors of point $v$ in the $k$-nn graph, and $\mathbf{m}_{uv}$ denotes as the message passing from the point $u$ to $v$. The GDL models typically need to capture geometric properties (*e.g.*, invariance properties), and this has caused GDL models to often process geometric features carefully. Beyond basic spatial coordinates, the GDL models that achieve the *invariance merit* often incorporate relative geometric information between points into the message $\mathbf{m}_{uv}$, such as distance [72], angle [25], torsion [51], and rotation angle [83] information. Note that the selected backbones in this study only involve the distance information. Investigation into how capturing higher-order geometric information (such as certain kinds of angles with special scientific meanings) affects the generalization ability of GDL models remains a topic for future work. Also, we refer interested readers to [16] for more detailed descriptions of different types of GDL models.

After several GDL layers, there is a pooling operator used to aggregate all point representations, to obtain the representation of the geometric data. Then an additional MLP component is needed to generate the predicted labels.

# C  Details of Datasets

## C.1  Dataset Statistics

The statistics of the covered datasets are shown in Table 4. Strategies of domain splits and sub-group splits for each distribution shift scenario, which have been discussed in Sec. 3.2, are detailed in Table

6. Note that for distribution shifts in DrugOOD-3D, we follow the same dataset splits and sub-group splits as the original benchmark DrugOOD. But in the Par-Label setting, we split 1000 samples from both Val-OOD and Test-OOD datasets, create the Train-OOD dataset, sample a specific number (100, 500, and 1000) for model fine-tuning, and evaluate the OOD performance of the fine-tuned models on the remaining OOD data. We also ensure a fair comparison here, as the number of removed samples is significantly smaller than the size of the OOD dataset itself. In the following three sub-sections, we make a complementary introduction to the studied scientific datasets.

Besides, we provide more granular information that better reflects the characteristics of the constructed datasets and distribution shifts. The detailed statistics can be seen in Table 5, covering

- The average number of tracks for each pileup level in Pileup Shift (Track-Pileup Dataset). Note that a higher PU level results in more background particle tracks in the collision while keeping the signal particle track the same.

- The average signal radius of each type of signal in Signal Shift (Track-Signal Dataset). Note that from $z \to 2\mu$, $z'_{20} \to 2\mu$, $z'_{10} \to 2\mu$, to $\tau \to 3\mu$, the average radius of signal tracks progressively approaches 2724.96 (the average radius of background tracks), which means it is getting harder to distinguish signals from backgrounds.

- The average number of atoms for different domains in Size Shift (DrugOOD-3D-Size Dataset).

- The average band gap value for each fidelity level in Fidelity Shift (QMOF Dataset). Note that the distinction between these fidelity levels extends beyond the mean of bandgap values. Specifically, the distribution of calculated band gap values displays varying properties across different levels, as illustrated in Fig. 1c.

Table 5: More granular information that better reflects the characteristics of the constructed datasets and distribution shifts.

| 1) **Pileup Shift** — the average number of tracks | | | |
|---|---|---|---|
| Domain | PU-10 | PU-50 | PU-90 |
| #Tracks | 55.76 | 232.58 | 408.38 |

| 2) **Signal Shift** — the average radius of signal tracks | | | |
|---|---|---|---|
| Domain | $\tau \to 3\mu$ | $z'_{10} \to 2\mu$ | $z'_{20} \to 2\mu$ | $z \to 2\mu$ |
| #Radius | 3979.66 | 8754.34 | 16014.98 | 58092.27 |

| 3) **Size Shift** — the average number of atoms | | | |
|---|---|---|---|
| Domain | Domain-8 | Domain-37 | Domain-95 | Domain-157 |
| #Atoms | 25 | 46 | 105 | 276 |

| 4) **Fidelity Shift** — the average bandgap value | | | |
|---|---|---|---|
| Domain | PBE | HLE17 | HSE06* | HSE06 |
| #Bandgap | 2.09 | 2.68 | 2.95 | 3.86 |

## C.2 Track Dataset

Here we employ the term *event* to refer to the comprehensive recording of an entire physics process by an experiment [74]. As mentioned in Sec. 3.2.1, a signal event (labeled as *positive*) involves the existence of a particular decay of interest (*i.e.*, signal). Here we are interested in multiple types of signals, including $z \to \mu\mu$, $\tau \to \mu\mu\mu$ (which have been widely observed), and $z'_K \to \mu\mu$ (which is a theoretical possibility) decays. This motivates us to construct the signal shift, where we expect the models trained with multiple types of signals to generalize to new signals that are different but to some extent related to the seen types. *Invariant mass* is a crucial physical quantity that characterizes the distinct decay type. Specifically, when ranked from the largest to the smallest, $z \to \mu\mu$ has an invariant mass of 91.19 GeV, $z'_K \to \mu\mu$ (where we consider $K = 80, 70, 60, 50$ for model training and $K = 10, 20$ for evaluation of model generalizability) has an invariant mass of $K$ GeV, and $\tau \to \mu\mu\mu$ has an invariant mass of 1.777 GeV. In our study, the disparities in invariant mass manifest through changes in the momenta of the signal particles and the radii of signal tracks (tracks left by signal particles). In the $z \to \mu\mu$ decay, the expected radius of the signal tracks is significantly

Table 6: The criteria of domain splits and sub-group splits in each distribution shift scenario. The "$\mathcal{S}$-Component" and "$\mathcal{T}$-Component" columns provide a description of the composition of the data in the domains $\mathcal{S}$ and $\mathcal{T}$. We denote the number of sub-group splits in the source domain $\mathcal{S}$ as |Sub-groups|. The criterion of the sub-group splits for each scenario is also summarized in the "Criterion" column.

| Dataset | Shift | Shift Case | $\mathcal{S}$-Component | |Sub-groups| | Criterion | $\mathcal{T}$-Component |
|---|---|---|---|---|---|---|
| Track-Pileup | Pileup | PU50 
 PU90 | PU10 | 5 | The number of points | PU50 
 PU90 |
| Track-Signal | Signal | $\tau \to 3\mu$ 
 $z'_{10} \to 2\mu$ 
 $z'_{20} \to 2\mu$ | Mixed Signals: $z \to 2\mu$ and $z'_K \to 2\mu$, where $K = 80, 70, 60, 50$, 5 types in total | 5 | The signal type | $\tau \to 3\mu$ 
 $z'_{10} \to 2\mu$ 
 $z'_{20} \to 2\mu$ |
| QMOF | Fidelity | HSE06 
 HSE06* | Mixed Fidelity: PBE, HLE17, HSE06* 
 Mixed Fidelity: PBE, HLE17, HSE06 | 3 
 3 | The fidelity level | HSE06 
 HSE06* |
| DrugOOD-3D-Assay | Assay | lbap-core-ic50-assay | | 307 | The assay environment | |
| DrugOOD-3D-Size | Size | lbap-core-ic50-size | Following DrugOOD | 91 | The molecular size | Following DrugOOD |
| DrugOOD-3D-Scaffold | Scaffold | lbap-core-ic50-scaffold | | 6682 | The scaffold pattern | |

larger, making it easily distinguishable from the background tracks, while in the $\tau \to \mu\mu\mu$ decay, the expected radius of the signal tracks is very close to that of the background tracks.

All events are simulated using the PYTHIA generator [6] with the addition of soft QCD pileup events, and particle tracks are generated using Acts [2]. Each point in a data entry is associated with a 3D coordinate, as well as other physical quantities measured by detectors, such as momenta. However, we use a dummy feature with all ones as the point feature for model training, following [55]. The model takes 3D coordinates and the dummy features of each point in data as input and predicts the existence of the signal in the given data.

### C.3 QMOF Dataset

We obtain 3D coordinates of each point in the materials data via the DFT-optimized structures provided by the QMOF Database. For point features, we associate each point in a sample with a categorical feature indicating the atom type for model training. The model takes 3D coordinates and atom-type categorical features as input and predicts the band gap value of given materials data.

### C.4 DrugOOD-3D Dataset

We first present how we adapt DrugOOD [35] and perform the GDL tasks over the dataset. We pre-process the SMILES [87] string of data provided in the dataset via the RDKit package [42], generating a conformer for each molecule, so as to assign each atom with a 3D coordinate. Concretely, we begin with generating a molecular object based on the SMILES string. Then we add hydrogens to the molecule and employ the ETKDG method [64] to obtain the initial conformer, which is further refined using the MMFF94 force field [29]. Note that we drop a data entry if it fails in conformer generation after the above process. The model takes 3D coordinates and atom-type categorical features as input, which is analogous to the scenario of the QMOF dataset, and predicts the binding affinity values of given ligands in a form of the binary classification task, as mentioned in Sec. 3.2.3.

### C.5 License

For the newly created Track-Pileup and Track-Signal datasets, we've got permission from the HEP community and utilized Acts to create them. Acts is licensed under the Mozilla Public License Version 2.0. Others are collected from public datasets and can be found at QMOF and DrugOOD. The data underlying the QMOF Database is made publicly available under a CC BY 4.0 license. DrugOOD is licensed under the GNU GENERAL PUBLIC LICENSE 3.0.

Also, note that the data we are using and curating do not contain any personally identifiable information or offensive content.

# D   Details of Algorithms and Backbones

## D.1   Backbone Details

Our benchmark contains **3** backbones which have been widely used in scenarios of geometric deep learning. Here we give detailed descriptions for each backbone in this study as follows.

- **DGCNN** (Dynamic Graph CNN), introduced by [85], is a GDL architecture aimed at exploiting local geometric structures of geometric data while maintaining permutation invariance. Specifically, it constructs a local neighborhood graph and applies edge convolution, with dynamic graph updates after each layer of the network.

- **Point Transformer** [110] is an architecture applying self-attention networks to 3D point cloud processing. It is built based on a highly expressive Point Transformer layer, which is invariant to permutation and cardinality of geometric data.

- **EGNN** ($E(n)$ Equivariant Graph Neural Networks), proposed by [71], is an architecture that preserves equivariance to rotations, translations and reflections on the coordinates of points when handling GDL data, *i.e.*, $E(n)$ equivariance, and that also preserves equivariance to permutations on the set of points.

**Discussions**. It is noteworthy that any specific application indicated by the datasets may have more advanced model architectures than these three architectures. We choose the above three as they are the most general, applicable to diverse scientific application scenarios, and most cutting-edge architectures are built upon them.

## D.2   Algorithm Details

Our benchmark contains **11** baselines spanning the No-Info, O-Feature, and Par-Label levels. We group them according to their distinct learning strategies and provide detailed descriptions for each algorithm as follows. We use ● to represent algorithms from the No-Info level, † for O-Feature, and ‡ for the Par-Label level, respectively.

- *Vanilla*: The empirical risk minimization (ERM) [82] minimizes the sum of errors across all samples.

- *Subgroup robustness*: Group distributionally robust optimization (GroupDRO) [68] aims to minimize worst-case losses and capture subgroup robustness by increasing the importance of groups with larger errors.

- *Invariant learning*: Variance Risk Extrapolation (VREx) [40] captures group invariance by specifically minimizing the risk variances of training domains.

- *Augmentation*: Mixup [105] improves model generalization by linearly interpolating two training samples randomly drawn from the training distribution. We follow [84] to perform Mixup specifically in the embedding space for the classification of geometric data.

- *Causal Inference*: DIR [91] captures the causal rationales for graph-structured data, mainly by conducting interventional augmentation on training data to create multiple interventional distributions, and then filtering out the parts of data that are unstable for model predictions. It is a well-known graph-based OOD baseline.

- *Information bottleneck*: LRI [55] is a novel geometric deep learning strategy grounded on a variational objective derived from the principle of information bottleneck. It injects learnable randomness to each node of geometric data, aimed at capturing minimal sufficient information to make correct and stable predictions. We adopt its *LRI-Bernoulli* framework, which specifically injects Bernoulli randomness to each point.

† *Domain Invariance for Unsupervised Domain Adaptation*: Domain-Adversarial Neural Network (DANN) [22] encourages feature representations to be consistent across the source and the target domain by adversarially training the normal label predictor and a special domain classifier; Deep correlation alignment (DeepCoral) [75] also encourages domain invariance by penalizing the deviation of covariance matrices between the source and the target domain.

‡ *Vanilla Fine-tuning*: We fine-tune all parameters of the GDL model using a small amount of OOD data, after it has been pre-trained on ID data via the ERM algorithm. Specifically, we conduct 3 baselines here, fine-tuning the model using 100, 500, and 1000 labeled target samples, respectively.

**Discussions**. Here we explain the rationale behind the selection of methods in this benchmark.

Firstly, the baselines need to include diverse learning strategies, as listed above. This means that if multiple methods fundamentally adopt similar ideas or strategies, we will select the most representative one among them.

Secondly, to our best knowledge, there are no OOD baselines specifically designed for scientific GDL, except LRI which assesses OOD generalization performance in its paper. To build this benchmark, it is necessary to extend general-purpose and foundational methodologies to the GDL setting. Therefore, the selected baselines cover 1) general-purpose methods, which can be applied to various scenarios such as CV, GraphML, and GDL; 2) graph-specific methods, applicable to GraphML but also feasible in GDL; and 3) GDL-specific methods, feasible only in GDL. That is why DIR and LRI are essential components of our selected baselines.

## E    Motivation and Insights from Experimental Design

Since we aim to understand how different Info levels and their associated algorithms affect the model generalizability across different application scenarios, it is important to carefully design experimental settings for a fair comparison among the three Info levels. As introduced in Sec 4.1,

- In the training stage, across the three levels, the model is trained (or pre-trained in Par-Label level) on the *same* amount of ID data, and gains additional access to some OOD features (in O-Feature level) or a few OOD features & labels (in Par-Label level). In this way, we demonstrate the effect of extra OOD info on model generalizability given that the factor of ID data is controlled invariant across the three levels.

- In the evaluation stage, across the three levels, we evaluate the model's ID performance using the *same* Val-ID and Test-ID datasets, and its OOD performance using the *same* Val-OOD and Test-OOD datasets, for a fair evaluation.

We consider the following cases: If additional OOD information does not improve generalization performance, it is unnecessary to consider the corresponding Info level. Conversely, if it enhances model generalizability, we need to further evaluate the practicality of utilizing this Info level and its associated algorithms by assessing: 1) the expected performance gain from the additional OOD information, and 2) the costs related to acquiring such info. Regarding point 1), the potential of extra OOD info to enhance generalization performance relies on the underlying mechanism of the distribution shift, as analyzed in Section 4.2 and 4.3. This requires evaluating the type and mechanism of the studied distribution shift using domain-specific knowledge. For point 2), we need to assess the availability and the cost of collecting some labeled or unlabeled OOD data in the considered application.

Accordingly, we recommend three steps to GDL practitioners in handling distribution shift issues: 1) Assess the type of shift by leveraging domain-specific knowledge; 2) Assess the availability of collecting some labeled or unlabeled OOD data; 3) Select the suitable Info level and its associated method by considering the trade-off between the expenses of gathering extra OOD information (dependent on step2) and the expected performance gain resulting from such info (dependent on step1). Therefore, our experimental design and results provide practitioners with insights for making the most suitable choice in handling scientific distribution shift issues.

## F    Details of Experimental Implementation

We conduct experiments on **3** scientific datasets and **10** cases of distribution shifts, covering **3** GDL backbones and **11** baselines from **3** knowledge levels. We implement our codes based on PyTorch Geometric [18]. We provide details of experimental implementation as follows.

**Basic Setup**. For all the experiments, we use the Adam optimizer, with a learning rate of 1e-3 and a weight decay of 1e-5. For each backbone, we use a fixed setting across various scenarios, all with

Table 7: Hyperparameters search space for all algorithms.

| Algorithm | Hyperparameter | Search Space |
|---|---|---|
| VREx | Penalty Weight | $\{0.001, 0.01, 0.1, 1.0\}$ |
| GroupDRO | Exponential Coefficient | $\{0.001, 0.01, 0.1, 1.0\}$ |
| Mixup | Probability | $\{0.25, 0.5, 0.75, 1.0\}$ |
| DIR | Causal Ratio | $\{0.3, 0.4, 0.5\}$ |
| LRI | Information Loss Coefficient | $\{0.01, 0.1, 1.0, 10.0\}$ |
| DANN | Domain Loss Weight | $\{0.001, 0.01, 0.1, 1.0, 5.0\}$ |
| DeepCoral | Penalty Weight | $\{0.001, 0.01, 0.1\}$ |

the sum global pooling and the RELU activation function. The settings of batch size, maximum number of epochs, and the number of iterations per epoch for the O-Feature level are consistent across different algorithms for a fair comparison in this study. Details are shown in Table 8. Note that the batch size is set to 128 instead of 256 in the O-Feature level of the pileup shift due to the memory constraints, and maximum number of epochs is set to 75 in the pileup shift because the model has been trained to converge under this setting.

Table 8: General hyperparameters of the datasets in this study.

| Dataset | Shift | No-Info | | O-Feature | | | Par-Label | |
|---|---|---|---|---|---|---|---|---|
| | | Batch Size | # Max Epochs | Batch Size | # Max Epochs | # Iterations per Epoch | Batch Size | # Max Epochs |
| Track | Pileup | 256 | 200 | 128 | 200 | 150 | 256 | 75 |
| | Signal | 256 | 100 | 256 | 100 | 150 | 256 | 100 |
| QMOF | Fidelity | 256 | 100 | 256 | 100 | 150 | 256 | 100 |
| DrugOOD-3D | Size, Scaffold, and Assay | 256 | 100 | 256 | 100 | 150 | 256 | 100 |

**Hyperparameter Tuning**. For each knowledge level and each algorithm, we search from a set of one specific hyperparameter to tune, and select the optimal one based on Val-OOD metric scores for a fair comparison. For VREx, we tune the weight of its variance penalty loss; For GroupDRO, we tune the Exponential coefficient; For Mixup, we tune the probability value that a certain batch data performs mixup augmentation; For DIR, we tune the causal ratio for selecting causal edges; For LRI, we tune the weight of the KL divergence regularizer; For DANN, we tune the weight of the domain classification loss; For DeepCoral, we tune the weight of covariance penalty loss. We detail the search space for each hyperparameter in Table 7.

## G   Complete Experimental Results

Here we present complementary baseline results that are not shown in the main text due to space limit in Table 9, 10, 11, and 12.

## H   Complementary Analysis of Experimental Results

In addition to representative analysis shown in the main text (*cf.* Sec 4.2 and 4.3), here we present complementary analysis (observations, conclusions, and conjectures) of experimental results to provide further insights to the community.

• **H1**. *Complementary to General Findings in Sec. 4.2*

We observe that OOD generalization methods in No-Info level find it hard to improve generalizability across various applications, which implies that the assumptions adopted by these methods may be kind of strong and not really match practical scenarios. Therefore, we recommend that future studies pay attention to 1) collecting some data information from the target domain $\mathcal{T}$ if possible, and 2) proposing novel OOD methods based on assumptions that better match the scientific applications.

• **H2**. *Complementary to Conclusion 1 in Sec. 4.3*

Based on the distinctions between Pileup and Signal shifts outlined in Conclusion 1, we propose two plausible explanations for the failure of DA strategies in Pileup Shift. One explanation lies in their distinct mechanisms (*cf.* Sec. 3.2.1): Pileup Shift represents $\mathcal{I}$-Conditional Shift, a shift occurring

exclusively in the independent part, *i.e.* $\mathbb{P}(X_i|X_c)$. Therefore, the OOD features, unlike in Signal Shift, cannot provide sufficient information to guide model predictions in the target domain.

Another conjecture is that, in Pileup Shift, the variation occurs in the number of tracks instead of geometric characteristics (such as the curvature of signal tracks in Signal Shift). Despite sensitive to geometric properties, the GDL model may struggle to handle the variation of such non-geometric information, which makes it more challenging to align the representation space between the source domain $\mathcal{S}$ and the target domain $\mathcal{T}$, when given the additional OOD feature info.

We are currently unable to control either of these two disparitiy factors for further exploration because it would destroy the inherent scientific implications of the `Track-Pileup` dataset. Therefore, we leave more investigation to future work.

• **H3**. *Complementary to Conclusion 1 in Sec. 4.3*

The DANN algorithm (and its corresponding O-Feature Level) shows a lower performance gain in the case of $\tau \to 3\mu$ compared to $z'_{10} \to 2\mu$ (*cf.* Fig. 4), although they are both categorized as $\mathcal{C}$-Conditional Shift. We explain this disparity by analyzing the geometric characteristics of the data causal component $X_c$. As shown in Table 5, the average radius of signal tracks in the $\tau \to 3\mu$ decay is very close to that of the background tracks, which means the signal and background tracks share very similar curvature in this case. Therefore, distinguishing signals from backgrounds is much more challenging in the case of $\tau \to 3\mu$, even when the model has access to the feature information of the $\tau \to 3\mu$ event in O-Feature Level.

• **H4**. *Complementary to Conclusion 2 in Sec. 4.3*

Additionally, we observe that TL strategies cannot yield a large improvement over ERM in Assay Shift (*cf.* Table 10), which is another scenario of Concept Shift (*cf.* Sec. 3.2.3). Following the analysis in Conclusion 2, a plausible explanation lies in how $\mathbb{P}_{\mathcal{S}}(Y)$ and $\mathbb{P}_{\mathcal{T}}(Y)$ varies in this case: Although there is a large divergence in $\mathbb{P}(Y)$ between different assay subgroups (*cf.* Fig. 6a), the distribution $\mathbb{P}(Y)$ is quite similar between the source and target domain (*cf.* Fig. 6b), *i.e.* $\mathbb{P}_{\mathcal{S}}(Y) \approx \mathbb{P}_{\mathcal{T}}(Y)$, which stands in contrast to the scenarios of the scaffold shift and size shift shown in Fig. 6c and 6d.

However, to provide a comprehensive answer to this question, it's crucial to consider other factors as well. For example, we simplify the affinity prediction by following DrugOOD, transitioning it to a binary classification task. Besides, the mechanism of the assay shift, unlike the fidelity shift scenario, may involve more than just a mismatch of label distribution $\mathbb{P}(Y)$ but could also entail a substantial shift in the input distribution $\mathbb{P}(X)$ between the domains $\mathcal{S}$ and $\mathcal{T}$. This can pose a significant challenge, particularly when the amount of labeled OOD data is limited. We leave a further in-depth analysis of this case to future work.

# I   Comparisons with Previous Findings

During result analysis, we identify conclusions that align with existing OOD literature and others that are novel and specific to the GDL setting. We conduct comparisons, including both consistency and disparity, between our observations and conclusions and those from previous findings related to OOD, such as in CV tasks.

## I.1   Comparison — Consistency

• In Sec. 4.2, we observe that fine-tuning can sometimes result in negative effects when the labeled OOD data is quite limited, particularly in cases involving a smaller degree of distribution shifts. This is consistent with [38], where fine-tuning a large model based on a small set of labels may lead to catastrophic forgetting. To mitigate this issue, robust fine-tuning strategies, such as weight-space ensembles [90], regularization [94] and surgical fine-tuning [44], could be potential solutions.

• In **H1** of Appendix H, we observe that multiple OOD generalization methods in our No-Info level find it hard to provide significant improvement across various applications. We can find consistent observations in existing works [14, 28, 39].

• In Sec. 4.3, we conclude that, "For the OOD generalization methods to learn robust representations, the more informatively the groups obtained by splitting the source domain $\mathcal{S}$ indicate the distribution

shift, the better performance these methods may achieve." This is consistent with [13], which revealed the importance of appropriate subgroup partitioning for invariant learning.

- Some works focus on leveraging additional auxiliary variables for OOD generalization. [92] used auxiliary information to help improve OOD performance in a semi-supervised scenario. [48] recently proposed to leverage such additional variables to encode information about the latent distribution shift, and to jointly learn group splits and invariant representation. How to leverage these auxiliary variables to enhance OOD generalization is an interesting topic for the GDL settings.

## I.2 Comparison — Disparity

- Conclusion 1 in Sec. 4.3 reveals how an algorithm can achieve superior performance by analyzing shifts in the geometric or non-geometric characteristics of the causal ($X_c$) or non-causal ($X_i$) components in geometric data. Such an analysis is tailored for GDL settings.

- The second point in Sec. I.1 could be even more severe in GDL compared to CV tasks considering the intricate nature of irregularity and geometric prior (information on the structure space and symmetry properties like invariance or equivariance) inherent in geometric data.

Besides, certain shifts in scientific GDL are infrequent or even unique in CV. This indicates the challenges faced by several methods initially proposed for CV tasks in addressing these shifts, and the necessity to develop OOD methods specifically designed for scientific GDL. We list some examples in our work as follows.

- Size shift, despite categorized as covariate shift, is a unique case where the model trained in data with lower size is to generalize to data with larger size. Methods designed for CV might struggle to capture this mechanism, potentially explaining why several methods do not perform well in the context of size shift in our study.

- Fidelity shift, which indicates the variation in the causal correlation between $X_c$ and $Y$, poses a challenge in material property prediction. However, such a shift in $\mathbb{P}(Y|X_c)$ is rare in CV tasks because $Y$ is typically derived from human annotations based on the input $X$. Therefore, most methods in our benchmark struggle to handle this type of shift, with the exception of the pretraining-finetuning strategy.

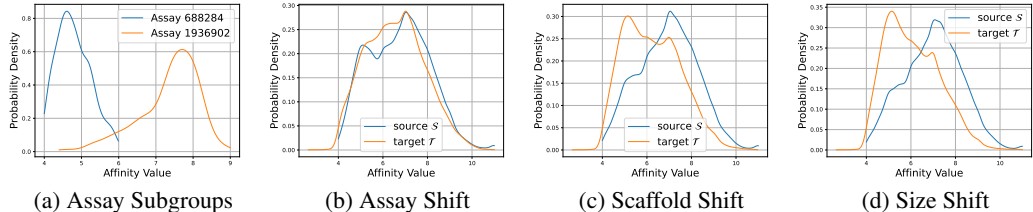

| (a) Assay Subgroups | (b) Assay Shift | (c) Scaffold Shift | (d) Size Shift |

Figure 6: Plotted KDE curves of the marginal label distribution $\mathbb{P}(Y)$. (a): $\mathbb{P}(Y)$ between two distinct assay subgroups, namely Assay 688284 and Assay 1936902, where $Y$ represents the ground-truth binding affinity value; (b) / (c) / (d): $\mathbb{P}_{\mathcal{S}}(Y)$ and $\mathbb{P}_{\mathcal{T}}(Y)$ in the assay / scaffold / size shift.

Table 9: Experimental results (Val-ID, Test-ID, Val-OOD, and Test-OOD performance included) on the $z'_{20} \to 2\mu$ case of the Signal shift over three backbones with the evaluation metrics of ACC (higher values indicate better performance). Note that the ID performance of TL methods is not evaluated. Parentheses show standard deviation across 3 replicates. We **bold** and underline the best and the second-best OOD performance for each distribution shift scenario.

| Level | Algorithm | EGNN | | | | DGCNN | | | | Pointtrans | | | |
|---|---|---|---|---|---|---|---|---|---|---|---|---|---|
| | | Val-ID | Test-ID | Val-OOD | Test-OOD | Val-ID | Test-ID | Val-OOD | Test-OOD | Val-ID | Test-ID | Val-OOD | Test-OOD |
| No-Info | ERM | 97.66(0.11) | 96.74(0.18) | 89.41(0.57) | 89.13(0.62) | 96.38(0.12) | 95.55(0.10) | 84.89(0.09) | 84.72(0.30) | 94.60(0.31) | 93.44(0.37) | 81.76(0.61) | 82.73(0.21) |
| | VREx | 97.57(0.12) | 96.75(0.24) | 88.93(0.71) | 89.02(0.71) | 96.16(0.16) | 95.43(0.44) | 84.22(0.29) | 84.17(0.12) | 94.18(0.10) | 93.18(0.34) | 82.54(0.03) | 83.42(0.41) |
| | GroupDRO | 97.62(0.13) | 96.74(0.23) | 89.80(0.37) | **89.55(0.08)** | 95.99(0.34) | 95.12(0.41) | 86.20(0.84) | 85.80(0.92) | 94.41(0.45) | 93.22(0.37) | 83.53(0.74) | 83.90(0.56) |
| | DIR | 94.57(1.81) | 94.12(1.88) | 84.57(2.44) | 84.86(2.62) | 92.98(1.18) | 91.78(0.95) | 83.78(1.25) | 83.98(1.08) | 85.84(10.21) | 84.92(9.97) | 76.39(4.88) | 77.34(5.29) |
| | LRI | 96.96(0.08) | 96.25(0.16) | 86.12(1.07) | 86.49(1.13) | 94.18(0.08) | 93.32(0.07) | 82.28(0.09) | 82.06(0.04) | 92.85(0.18) | 91.75(0.13) | 80.58(0.83) | 81.21(0.41) |
| | MixUp | 97.76(0.04) | 97.05(0.24) | 89.18(0.16) | 88.86(0.41) | 96.48(0.14) | 95.43(0.12) | 85.28(0.58) | 85.15(0.88) | 94.44(0.08) | 93.31(0.05) | 81.91(0.67) | 82.83(0.79) |
| O-Feature | DANN | 96.13(0.20) | 95.16(0.68) | 89.54(0.31) | 89.49(0.31) | 94.99(0.48) | 94.26(0.19) | 88.51(0.21) | **88.33(0.20)** | 90.81(0.26) | 90.13(0.16) | 82.98(0.57) | 83.15(0.50) |
| | Coral | 97.71(0.21) | 96.92(0.20) | 88.86(0.01) | 89.07(0.26) | 95.28(0.11) | 94.54(0.17) | 84.33(0.50) | 84.54(0.59) | 94.17(0.18) | 93.23(0.11) | 81.58(1.23) | 82.50(0.82) |
| Par-Label | TL$_{100}$ | | | 78.33(0.35) | 78.33(0.99) | | | 73.45(0.98) | 71.13(1.25) | | | 82.80(0.80) | 82.98(0.86) |
| | TL$_{500}$ | | | 82.09(0.68) | 82.81(0.27) | | | 79.02(1.52) | 78.31(1.83) | | | 83.68(0.23) | 84.08(0.38) |
| | TL$_{1000}$ | | | 84.33(0.23) | 84.90(0.40) | | | 80.67(1.15) | 79.80(1.52) | | | 84.42(0.38) | **84.59(0.40)** |

Table 10: Experimental results on the **Assay** shift over three backbones, with the evaluation metrics of AUC (higher values indicate better performance). Note that the ID performance of TL methods is not evaluated. Parentheses show standard deviation across 3 replicates. We **bold** and underline the best and the second-best OOD performance for each distribution shift scenario.

| Level | Algorithm | EGNN | | | | DGCNN | | | | Pointtrans | | | |
|---|---|---|---|---|---|---|---|---|---|---|---|---|---|
| | | Val-ID | Test-ID | Val-OOD | Test-OOD | Val-ID | Test-ID | Val-OOD | Test-OOD | Val-ID | Test-ID | Val-OOD | Test-OOD |
| No-Info | ERM | 92.35(0.07) | 91.70(0.11) | 70.66(0.03) | 70.85(0.65) | 90.49(0.06) | 90.07(0.03) | 71.44(0.33) | 70.77(0.34) | 89.54(0.12) | 89.31(0.07) | 70.55(0.36) | 69.58(0.42) |
| | VREx | 92.08(0.09) | 91.67(0.13) | 72.59(1.05) | 71.21(0.47) | 89.81(0.23) | 89.38(0.09) | 71.40(0.16) | 70.72(0.20) | 89.53(0.09) | 89.23(0.07) | 70.25(0.25) | 69.85(0.38) |
| | GroupDRO | 92.05(0.10) | 91.21(0.04) | 72.15(0.24) | **71.82(0.71)** | 88.79(0.16) | 88.45(0.24) | 71.62(0.06) | **71.69(0.70)** | 87.93(0.15) | 87.87(0.15) | 69.94(0.16) | 70.37(0.30) |
| | DIR | 82.57(1.66) | 81.85(1.94) | 70.08(0.98) | 67.97(2.15) | 84.25(0.57) | 83.89(0.52) | 69.91(0.41) | 68.55(1.24) | 86.14(1.09) | 85.99(1.05) | 68.79(0.40) | 68.20(0.23) |
| | LRI | 92.20(0.07) | 91.31(0.15) | 71.31(0.58) | 70.41(0.13) | 90.67(0.09) | 90.13(0.09) | 71.03(0.09) | 70.93(0.34) | 89.28(0.08) | 89.11(0.18) | 69.80(0.11) | 69.83(0.50) |
| | MixUp | 92.25(0.14) | 91.55(0.22) | 71.15(0.18) | 71.36(0.21) | 90.53(0.01) | 90.06(0.12) | 70.88(0.33) | 70.71(0.24) | 89.46(0.14) | 89.20(0.10) | 70.00(0.25) | **70.65(0.41)** |
| O-Feature | DANN | 91.00(0.09) | 90.39(0.07) | 72.13(1.47) | 71.76(0.87) | 90.56(0.13) | 90.28(0.16) | 70.47(0.29) | 70.31(0.42) | 89.65(0.21) | 89.33(0.13) | 69.90(0.22) | 69.71(0.17) |
| | Coral | 92.38(0.05) | 91.84(0.25) | 71.51(0.66) | 71.29(0.55) | 90.59(0.16) | 90.03(0.17) | 70.80(0.55) | 70.14(0.97) | 89.66(0.21) | 89.39(0.15) | 70.02(0.47) | 69.51(0.68) |
| Par-Label | TL$_{100}$ | | | 68.73(0.98) | 68.82(0.47) | | | 67.27(0.53) | 69.14(0.74) | | | 69.31(0.53) | 69.77(0.14) |
| | TL$_{500}$ | | | 70.41(0.30) | 70.81(0.70) | | | 69.01(0.51) | 69.83(0.63) | | | 69.70(0.47) | 70.02(0.28) |
| | TL$_{1000}$ | | | 73.66(1.18) | 71.44(0.49) | | | 70.95(0.53) | 71.19(0.34) | | | 69.61(0.59) | 70.30(0.14) |

Table 11: Experimental results (Val-ID and Val-OOD performance) on **Pileup** (PU50 and PU90 cases), **Signal** ($\tau \to 3\mu$ and $z'_{10} \to 2\mu$ cases), **Size**, **Scaffold**, and **Fidelity** (HSE06 and HSE06* cases) shifts over the backbones of EGNN and DGCNN. Note that Val-ID performance of TL methods is not evaluated. Parentheses show standard deviation across 3 replicates. ↑ denotes higher values correspond to better performance, whereas ↓ denotes lower for better. We **bold** and underline the best and the second-best OOD performance for each distribution shift scenario.

**Pileup Shift — $\mathcal{I}$-Conditional Shift (ACC↑)**

| Level | Algorithm | EGNN | | | | DGCNN | | | |
|---|---|---|---|---|---|---|---|---|---|
| | | PU50 | | PU90 | | PU50 | | PU90 | |
| | | Val-ID | Val-OOD | Val-ID | Val-OOD | Val-ID | Val-OOD | Val-ID | Val-OOD |
| No-Info | ERM | 96.18(0.05) | 88.68(0.31) | 96.29(0.22) | 82.76(0.88) | 94.66(0.43) | 86.49(1.10) | 94.66(0.43) | 79.63(1.48) |
| | VREx | 96.05(0.12) | 88.36(0.43) | 96.05(0.12) | 81.63(0.73) | 94.95(0.22) | **86.92(0.59)** | 94.95(0.22) | **80.65(0.93)** |
| | GroupDRO | 93.09(0.36) | 83.82(0.33) | 93.09(0.36) | 76.71(0.19) | 91.79(0.16) | 79.94(0.36) | 91.79(0.16) | 74.45(0.21) |
| | DIR | 95.50(0.19) | 86.57(0.40) | 95.50(0.19) | 79.74(0.31) | 94.44(0.18) | 84.48(0.46) | 94.44(0.18) | 76.43(1.19) |
| | LRI | 96.28(0.09) | 88.88(0.27) | 95.89(0.23) | 82.80(0.71) | 94.52(0.13) | 86.08(0.09) | 94.52(0.13) | 79.61(0.62) |
| | MixUp | 96.25(0.10) | **89.29(0.24)** | 96.25(0.10) | 82.67(0.41) | 94.86(0.38) | 86.29(0.46) | 94.86(0.38) | 80.42(0.66) |
| O-Feature | DANN | 95.53(0.39) | 87.60(0.86) | 95.96(0.11) | 82.16(0.83) | 94.35(0.21) | 85.29(0.56) | 94.69(0.43) | 76.87(1.69) |
| | Coral | 95.69(0.23) | 87.65(0.87) | 95.88(0.20) | 79.73(2.43) | 94.46(0.28) | 84.97(1.04) | 94.91(0.42) | 78.24(1.08) |
| Par-Label | TL$_{100}$ | | 85.79(0.53) | | 80.31(0.70) | | 81.79(0.94) | | 74.13(0.38) |
| | TL$_{500}$ | | 87.76(0.27) | | 83.31(0.96) | | 84.91(1.09) | | 79.00(1.17) |
| | TL$_{1000}$ | | 88.57(0.19) | | **84.87(0.53)** | | 85.19(0.86) | | 79.94(0.20) |

**Signal Shift — $\mathcal{C}$-Conditional Shift (ACC↑)**

| Level | Algorithm | EGNN | | | | DGCNN | | | |
|---|---|---|---|---|---|---|---|---|---|
| | | $\tau \to 3\mu$ | | $z'_{10} \to 2\mu$ | | $\tau \to 3\mu$ | | $z'_{10} \to 2\mu$ | |
| | | Val-ID | Val-OOD | Val-ID | Val-OOD | Val-ID | Val-OOD | Val-ID | Val-OOD |
| No-Info | ERM | 97.85(0.12) | 67.11(0.49) | 97.72(0.04) | 71.30(0.78) | 96.55(0.15) | 66.12(0.41) | 96.47(0.07) | 69.71(0.20) |
| | VREx | 97.71(0.18) | 67.51(0.66) | 97.51(0.12) | 72.08(0.52) | 96.30(0.14) | 66.13(0.19) | 96.42(0.22) | 69.53(0.55) |
| | GroupDRO | 97.38(0.17) | 67.92(0.17) | 97.68(0.24) | 73.25(0.34) | 95.81(0.17) | 67.20(0.42) | 95.75(0.24) | 71.15(0.38) |
| | DIR | 77.80(2.78) | 68.47(0.13) | 93.94(2.86) | 71.18(0.91) | 92.91(0.47) | 66.21(0.64) | 92.91(0.47) | 71.45(0.42) |
| | LRI | 96.96(0.08) | 68.34(0.28) | 96.95(0.12) | 70.76(0.72) | 91.31(0.69) | 68.59(0.08) | 94.32(0.38) | 68.91(0.08) |
| | MixUp | 97.74(0.30) | 66.55(1.30) | 97.83(0.08) | 71.37(1.24) | 96.25(0.16) | 66.23(1.22) | 96.48(0.22) | 69.99(0.28) |
| O-Feature | DANN | 82.06(1.10) | **69.45(0.12)** | 91.20(1.04) | **77.15(0.45)** | 81.37(1.60) | **69.17(0.02)** | 88.97(0.29) | **75.61(0.16)** |
| | Coral | 96.94(0.76) | 67.96(0.65) | 97.70(0.08) | 71.97(0.26) | 95.52(0.08) | 65.91(0.76) | 95.17(0.15) | 69.88(0.76) |
| Par-Label | TL$_{100}$ | | 64.15(1.10) | | 68.69(0.84) | | 65.57(0.80) | | 65.07(1.10) |
| | TL$_{500}$ | | 67.11(1.04) | | 71.21(0.86) | | 67.75(0.93) | | 67.73(0.63) |
| | TL$_{1000}$ | | 68.96(0.67) | | 72.79(0.22) | | 68.56(0.10) | | 68.47(0.86) |

**Size & Scaffold Shift — Covariate Shift (AUC↑)**

| Level | Algorithm | EGNN | | | | DGCNN | | | |
|---|---|---|---|---|---|---|---|---|---|
| | | Size | | Scaffold | | Size | | Scaffold | |
| | | Val-ID | Val-OOD | Val-ID | Val-OOD | Val-ID | Val-OOD | Val-ID | Val-OOD |
| No-Info | ERM | 91.83(0.21) | 78.96(0.07) | 94.16(0.19) | 75.89(0.78) | 90.32(0.03) | 77.04(0.10) | 91.16(0.10) | 75.98(0.22) |
| | VREx | 91.56(0.23) | 78.94(0.42) | 94.41(0.36) | 76.41(1.04) | 90.07(0.12) | 77.47(0.27) | 91.85(0.74) | 76.63(0.47) |
| | GroupDRO | 87.46(0.28) | 74.08(0.27) | 94.22(0.26) | 76.77(0.66) | 83.99(0.24) | 73.05(0.12) | 91.56(0.19) | **76.79(0.13)** |
| | DIR | 87.83(1.03) | 75.57(0.46) | 89.23(2.45) | 73.46(2.61) | 80.99(0.32) | 72.03(0.63) | 79.66(1.19) | 73.09(0.80) |
| | LRI | 91.85(0.22) | **79.24(0.29)** | 94.35(0.22) | 76.38(0.09) | 90.27(0.38) | 77.23(0.14) | 87.83(0.16) | 75.82(0.31) |
| | MixUp | 91.70(0.27) | 79.05(0.23) | 94.09(0.24) | 77.32(0.15) | 90.24(0.14) | 77.35(0.24) | 91.90(0.09) | **76.81(0.32)** |
| O-Feature | DANN | 91.98(0.15) | 79.07(0.12) | 94.88(0.12) | 76.65(0.14) | 89.79(0.17) | 77.04(0.17) | 91.59(0.51) | 75.70(0.26) |
| | Coral | 92.07(0.20) | 79.01(0.44) | 95.15(0.18) | 76.81(0.29) | 89.68(0.20) | **77.62(0.22)** | 91.89(0.50) | 75.40(0.25) |
| Par-Label | TL$_{100}$ | | 77.53(0.69) | | 74.90(0.70) | | 76.49(0.26) | | 74.99(0.57) |
| | TL$_{500}$ | | 77.80(0.50) | | 77.12(0.83) | | 76.59(0.22) | | 76.18(0.27) |
| | TL$_{1000}$ | | 77.99(0.18) | | **77.64(0.26)** | | 76.57(0.11) | | 76.45(0.27) |

**Fidelity Shift — Concept Shift (MAE↓)**

| Level | Algorithm | EGNN | | | | DGCNN | | | |
|---|---|---|---|---|---|---|---|---|---|
| | | HSE06 | | HSE06* | | HSE06 | | HSE06* | |
| | | Val-ID | Val-OOD | Val-ID | Val-OOD | Val-ID | Val-OOD | Val-ID | Val-OOD |
| No-Info | ERM | 0.498(0.006) | 1.128(0.094) | 0.618(0.005) | 0.541(0.007) | 0.486(0.005) | 1.126(0.032) | 0.601(0.004) | 0.537(0.009) |
| | VREx | 0.498(0.005) | 1.110(0.068) | 0.619(0.004) | **0.524(0.013)** | 0.508(0.003) | 1.060(0.089) | 0.619(0.003) | 0.520(0.009) |
| | GroupDRO | 0.530(0.000) | 1.029(0.029) | 0.674(0.003) | **0.525(0.004)** | 0.512(0.005) | 1.012(0.017) | 0.684(0.007) | **0.505(0.003)** |
| O-Feature | DANN | 0.495(0.002) | 1.185(0.017) | 0.620(0.001) | 0.542(0.010) | 0.484(0.004) | 1.093(0.033) | 0.603(0.003) | 0.534(0.007) |
| | Coral | 0.499(0.007) | 1.182(0.044) | 0.618(0.005) | 0.554(0.006) | 0.489(0.001) | 1.100(0.017) | 0.603(0.002) | 0.526(0.010) |
| Par-Label | TL$_{100}$ | | 0.726(0.010) | | 0.606(0.028) | | 0.702(0.019) | | 0.583(0.003) |
| | TL$_{500}$ | | 0.640(0.008) | | 0.543(0.016) | | 0.625(0.005) | | 0.524(0.006) |
| | TL$_{1000}$ | | **0.619(0.007)** | | 0.535(0.012) | | **0.586(0.004)** | | 0.508(0.002) |

Table 12: Experimental results (Val-ID, Test-ID, Val-OOD, and Test-OOD performance included) on **Pileup** (PU50 and PU90 cases), **Signal** ($\tau \to 3\mu$ and $z'_{10} \to 2\mu$ cases), **Size**, **Scaffold**, and **Fidelity** (HSE06 and HSE06* cases) shifts over the backbone of **Point Transformer**. Note that the ID performance of TL methods is not evaluated. Parentheses show standard deviation across 3 replicates. ↑ denotes higher values correspond to better performance, whereas ↓ denotes lower for better. We **bold** and underline the best and the second-best OOD performance for each distribution shift scenario.

| | | **Pileup Shift — $\mathcal{I}$-Conditional Shift (ACC↑)** | | | | | | | |
|---|---|---|---|---|---|---|---|---|---|
| | | PU50 | | | | PU90 | | | |
| Level | Algorithm | Val-ID | Test-ID | Val-OOD | Test-OOD | Val-ID | Test-ID | Val-OOD | Test-OOD |
| No-Info | ERM | 93.93(0.36) | 93.15(0.31) | 85.25(0.25) | 84.07(0.60) | 93.93(0.36) | 93.15(0.31) | 79.73(0.10) | 78.67(0.25) |
| | VREx | 93.74(0.42) | 93.17(0.33) | 84.95(0.66) | 83.75(0.29) | 93.74(0.42) | 93.17(0.33) | 79.41(0.39) | 77.92(0.19) |
| | GroupDRO | 92.27(0.30) | 91.59(0.21) | 82.49(0.75) | 81.35(0.78) | 92.27(0.30) | 91.59(0.21) | 74.45(1.52) | 73.66(1.49) |
| | DIR | 93.15(0.13) | 92.81(0.14) | 84.79(0.64) | 84.13(0.45) | 93.15(0.13) | 92.81(0.14) | 79.71(0.83) | 78.92(0.76) |
| | LRI | 93.58(0.20) | 92.96(0.27) | 83.93(0.27) | 83.63(0.63) | 93.58(0.20) | 92.96(0.27) | 78.77(0.41) | 77.77(0.50) |
| | MixUp | 93.79(0.12) | 93.16(0.24) | 85.41(0.24) | **84.55(0.59)** | 93.79(0.12) | 93.16(0.24) | 80.17(0.34) | **79.15(0.46)** |
| O-Feature | DANN | 93.82(0.13) | 93.01(0.14) | 85.06(0.34) | 84.25(0.60) | 93.75(0.21) | 92.84(0.31) | 78.22(0.81) | 77.11(0.89) |
| | Coral | 93.45(0.07) | 92.88(0.14) | 84.87(0.10) | 83.97(0.12) | 93.59(0.12) | 92.88(0.12) | 77.51(1.20) | 76.38(1.73) |
| Par-Label | TL$_{100}$ | | | 83.39(0.38) | 82.76(0.30) | | | 77.95(1.62) | 76.59(1.88) |
| | TL$_{500}$ | | | 84.59(0.07) | 83.54(0.45) | | | 79.19(0.43) | 78.28(0.70) |
| | TL$_{1000}$ | | | 84.82(0.18) | 84.32(0.27) | | | 79.89(0.13) | 78.64(0.33) |

| | | **Signal Shift — $\mathcal{C}$-Conditional Shift (ACC↑)** | | | | | | | |
|---|---|---|---|---|---|---|---|---|---|
| | | $\tau \to 3\mu$ | | | | $z'_{10} \to 2\mu$ | | | |
| Level | Algorithm | Val-ID | Test-ID | Val-OOD | Test-OOD | Val-ID | Test-ID | Val-OOD | Test-OOD |
| No-Info | ERM | 94.60(0.31) | 93.44(0.37) | 68.66(0.14) | 67.43(0.13) | 94.60(0.31) | 93.44(0.37) | 68.25(0.39) | 66.96(0.92) |
| | VREx | 94.29(0.25) | 93.24(0.15) | 68.68(0.21) | 67.63(0.08) | 94.29(0.25) | 93.24(0.15) | 69.88(0.23) | 68.28(0.40) |
| | GroupDRO | 94.22(0.40) | 92.98(0.52) | 67.65(0.32) | 66.46(0.43) | 94.22(0.40) | 92.98(0.52) | 70.60(0.61) | 69.15(0.89) |
| | DIR | 85.84(10.21) | 84.92(9.97) | 68.16(0.55) | 67.31(0.49) | 85.84(10.21) | 84.92(9.97) | 68.59(1.93) | 66.64(1.81) |
| | LRI | 92.85(0.18) | 91.75(0.13) | 68.74(0.02) | 67.55(0.08) | 92.85(0.18) | 91.75(0.13) | 69.56(0.30) | 68.22(0.89) |
| | MixUp | 94.51(0.10) | 93.47(0.24) | 68.63(0.16) | 67.58(0.06) | 94.44(0.08) | 93.31(0.05) | 69.07(0.88) | 67.41(1.29) |
| O-Feature | DANN | 94.53(0.30) | 93.20(0.14) | 68.69(0.04) | **67.64(0.09)** | 83.97(0.30) | 84.07(0.35) | 72.26(0.10) | **70.75(0.35)** |
| | Coral | 94.42(0.24) | 93.32(0.25) | 68.71(0.05) | 67.60(0.05) | 94.61(0.27) | 93.44(0.29) | 68.39(0.98) | 67.47(0.92) |
| Par-Label | TL$_{100}$ | | | 66.39(1.66) | 65.87(1.20) | | | 69.57(1.63) | 68.92(1.96) |
| | TL$_{500}$ | | | 68.52(0.26) | 67.60(0.23) | | | 68.80(1.21) | 67.69(1.08) |
| | TL$_{1000}$ | | | 68.63(0.13) | 67.32(0.10) | | | 70.82(2.03) | 69.81(2.00) |

| | | **Size & Scaffold Shift — Covariate Shift (AUC↑)** | | | | | | | |
|---|---|---|---|---|---|---|---|---|---|
| | | Size | | | | Scaffold | | | |
| Level | Algorithm | Val-ID | Test-ID | Val-OOD | Test-OOD | Val-ID | Test-ID | Val-OOD | Test-OOD |
| No-Info | ERM | 88.91(0.28) | 88.09(0.58) | 76.34(0.25) | 64.17(0.49) | 90.05(0.25) | 81.22(0.42) | 75.26(0.61) | 67.92(0.46) |
| | VREx | 88.44(0.28) | 87.90(0.35) | 76.30(0.16) | 64.44(0.34) | 89.35(0.31) | 80.96(0.33) | 75.20(0.19) | 67.97(0.47) |
| | GroupDRO | 83.52(0.20) | 82.71(0.37) | 71.89(0.37) | 58.19(0.46) | 89.29(0.67) | 81.05(0.29) | 75.32(0.25) | 67.93(0.27) |
| | DIR | 83.65(2.49) | 83.46(2.40) | 73.63(1.14) | 62.82(0.91) | 83.61(3.02) | 77.05(1.05) | 72.11(0.67) | 65.82(1.11) |
| | LRI | 88.34(0.58) | 87.70(0.77) | 76.35(0.24) | 64.43(0.45) | 85.70(0.27) | 79.08(0.21) | 74.15(0.18) | 67.34(0.15) |
| | MixUp | 88.76(0.07) | 88.17(0.21) | 76.58(0.13) | 63.81(0.13) | 89.40(0.46) | 80.88(0.22) | 75.00(0.07) | 67.56(0.20) |
| O-Feature | DANN | 88.13(0.12) | 87.61(0.07) | 76.12(0.16) | 64.76(0.33) | 89.87(0.16) | 80.70(0.20) | 74.30(0.24) | 67.26(0.31) |
| | Coral | 88.33(0.70) | 87.91(0.38) | 76.60(0.15) | 64.57(0.12) | 90.26(0.47) | 80.44(0.40) | 74.49(0.69) | 67.45(0.36) |
| Par-Label | TL$_{100}$ | | | 75.90(0.20) | 64.11(0.38) | | | 74.07(0.56) | 67.67(0.09) |
| | TL$_{500}$ | | | 75.97(0.37) | 64.33(0.47) | | | 75.20(0.52) | 68.35(0.18) |
| | TL$_{1000}$ | | | 75.89(0.36) | **65.14(0.90)** | | | 76.32(0.41) | **70.00(0.15)** |

| | | **Fidelity Shift — Concept Shift (MAE↓)** | | | | | | | |
|---|---|---|---|---|---|---|---|---|---|
| | | HSE06 | | | | HSE06* | | | |
| Level | Algorithm | Val-ID | Test-ID | Val-OOD | Test-OOD | Val-ID | Test-ID | Val-OOD | Test-OOD |
| No-Info | ERM | 0.492(0.002) | 0.495(0.002) | 1.182(0.014) | 1.146(0.014) | 0.613(0.003) | 0.624(0.007) | 0.543(0.002) | 0.553(0.003) |
| | VREx | 0.522(0.009) | 0.517(0.008) | 1.102(0.044) | 1.080(0.033) | 0.621(0.003) | 0.623(0.004) | 0.523(0.002) | 0.536(0.004) |
| | GroupDRO | 0.527(0.005) | 0.516(0.008) | 0.993(0.052) | 0.959(0.057) | 0.641(0.015) | 0.643(0.016) | 0.513(0.004) | 0.529(0.003) |
| O-Feature | DANN | 0.493(0.001) | 0.501(0.003) | 1.162(0.033) | 1.135(0.038) | 0.612(0.001) | 0.615(0.004) | 0.537(0.007) | 0.560(0.011) |
| | Coral | 0.491(0.003) | 0.498(0.000) | 1.212(0.027) | 1.181(0.033) | 0.612(0.006) | 0.618(0.013) | 0.541(0.006) | 0.561(0.003) |
| Par-Label | TL$_{100}$ | | | 0.684(0.010) | 0.689(0.015) | | | 0.583(0.008) | 0.598(0.006) |
| | TL$_{500}$ | | | 0.618(0.008) | 0.613(0.005) | | | 0.519(0.005) | 0.545(0.006) |
| | TL$_{1000}$ | | | 0.583(0.001) | **0.584(0.002)** | | | 0.511(0.007) | **0.522(0.010)** |

