# OpenReview forum: "GeSS: Benchmarking Geometric Deep Learning under Scientific Applications with Distribution Shifts"
_NeurIPS.cc/2024/Datasets_and_Benchmarks_Track — NeurIPS 2024 Track Datasets and Benchmarks Poster_

### Official Review · Reviewer_xiTX · 2024-06-24
**OOD Benchmark data under different scientific applications.**

**Rating:** 7
**Confidence:** 3
**Correctness:** The claims are correct.
**Clarity:** The paper is well written.

**Review:**

Pro:
1. The Domain Shift problem is now detailed across three specific categories, enhancing the analysis.
2. The paper addresses various topics in different scientific applications, highlighting common issues in GDL applications.
3. The inclusion of 3 backbones and 11 learning algorithms makes the study very comprehensive.
4. The in-depth analysis provided can benefit future researchers.

Con:

1. The paper contains numerous acronyms, which can be confusing for readers who are not well-versed in the topic, like myself. It would be helpful to include a table of acronyms or provide explanations for them.

**Strengths:**

I find this paper to be very comprehensive, providing valuable insights for future research on the OOD problem.

**Additional Feedback:**

I don't have any additional feedback.

**Documentation:**

The data collection is in detail.

**Ethics:**

There are no ethical concerns.

**Limitations:**

The limitations are described.

**Opportunities For Improvement:**

It's a pretty good paper. I don't have further suggestions.

**Relation To Prior Work:**

Yes. Section 2 is about this.

**Summary And Contributions:**

The authors introduce a benchmark dataset named GeSS, encompassing particle physics, materials science, and biochemistry, to explore the out-of-distribution (OOD) problem. They define the OOD problem through three scenarios: no OOD information, only unlabeled OOD data, and OOD data with a few labels. They test three geometric deep learning (GDL) backbones—EGNN, DGCNN, and Point Transformer—using 11 different learning algorithms. The comprehensive analysis provides insights for addressing domain shifts in the future.

---

> ### Author Rebuttal · Authors · 2024-08-15
>
> We express our gratitude to the reviewer xiTX for acknowledging the contribution of our benchmark and recognizing the detailed problem formulation, broad range of cases for GDL in ML4Science, comprehensive evaluation, and in-depth analysis of this work. We also value the advice on including a table of acronyms and give corresponding explanations. We provide an acronyms table in the rebuttal and will include this table with detailed explanations and references to the final version. Also, if your concerns are satisfactorily addressed, we would be grateful if you could consider improving the rating of our work.
>
> | acronyms  |                     Meanings                      |
> | :-------: | :-----------------------------------------------: |
> |    GDL    |         Geometric Deep Learning, Abstract         |
> |    OOD    |           Out-of-Distribution, Abstract           |
> |   ML4S    |        Machine Learning for Science, Intro        |
> |    HEP    |            High Energy Physics, Intro             |
> |  No-Info  |        Levels of no OOD information, Intro        |
> | O-Feature | Levels of only OOD features without labels, Intro |
> | Par-Label |  Levels of OOD features with a few labels, Intro  |
> |    DA     |         Domain Adaptation methods, Intro          |
> |    TL     |         Transfer Learning methods, Intro          |
> |  GraphML  | Machine Learning on Graph-structured data, Sec 2  |
> |    PU     |              PileUp level, Sec 3.2.1              |
> |   MOFs    |        Metal-Organic Frameworks, Sec 3.2.2        |
> |    DFT    |       Density Functional Theory, Sec 3.2.2        |
> |   LBAP    |    Ligand Based Affinity Prediction, Sec 3.2.3    |
> |    SCM    |       Structure Causal Model, Appendix A.1        |

---

> > ### Comment · Reviewer_xiTX · 2024-08-29
> >
> > The acronym table effectively addresses my concern, providing clarity and improving the paper’s overall presentation. I find the paper to be well-executed and informative. With no further questions, I will maintain my score at 7.

---

> > > ### Author Response · Authors · 2024-08-29
> > > **Thanks for the feedback**
> > >
> > > Many thanks for checking our response. Thank you for your support!
> > >
> > > the authors

---

### Official Review · Reviewer_UETZ · 2024-07-24
**Initial review**

**Rating:** 6
**Confidence:** 3
**Correctness:** Yes.
**Clarity:** Yes.

**Review:**

**Quality:** The overall quality of this paper is good. The paper is well-written, and the experiments are comprehensive.

**Clarity:** This paper is well-clarified. The motivations of the proposed dataset and benchmark are well-defined. A comparison between the proposed benchmark and the existing benchmarks is also provided. However, a datasheet is missing in this paper.

**Originality:** This paper presents a novel dataset and benchmark for GDL with distribution shift.

**Significance:** The dataset is an important complementary to the current benchmarks on distribution shift.

**Strengths:**

- This paper presents a new dataset and benchmark in the context of GDL with distribution shift.

- This paper is well-written and well-motivated.

- The experiments are comprehensive. Testing examples from both simulation and experiments are appreciated.

**Additional Feedback:**

- Are there any domain-specific metrics for evaluating the performance of baseline models in Table 2? It seems many of them are commonly-used metrics. For material science, are there any energy-based evaluations?

- Do the authors observe the similar performance of OOD methods across different scientific datasets?

**Documentation:**

No. A datasheet is missing.

**Ethics:**

No.

**Limitations:**

Yes.

**Opportunities For Improvement:**

- It seems that a datasheet is missing from this paper. Including this would greatly enhance the completeness of the work.

- It would be better to include some more details of the simulation and experimental details of the datastes. It also would be better to include the motivations of using the selected GDL baseline models.

**Relation To Prior Work:**

Yes.

**Summary And Contributions:**

This paper presents a benchmark for testing geometric deep learning methods in scientific scenarios with distribution shifts. The datasets include various scientific domains and consider multiple different types of shifts. Both simulation data and experimental data are provided. Various experiments have been conducted to test the baseline performances.

---

> ### Author Rebuttal · Authors · 2024-08-15
>
> We thank the reviewer UETZ for recognizing the well-clarified motivation, well-written paper, and comprehensive experiments of this work. We summarize the reviewer’s concerns and questions and provide our explanations as follows. If your concerns are satisfactorily addressed, we would be grateful if you could consider improving the rating of our work.
> ## 1. Opportunities For Improvement
> Thank you for your valuable suggestions. We appreciate the reviewer’s attention to details and agree that these aspects are crucial for the completeness of the work. We would like to kindly point out that some details the reviewer asked for were put in the appendix of the initial manuscript due to the limited space of the main text. We apologize if this was not clear due to the placement in the appendix. We will use LaTeX boxes to make these more prominently referenced in the final version.
> ### 1.1 A datasheet is missing
> We have included a detailed datasheet in Table 4,5,6 of Appendix C (page 20-22). Table 4 shows Dataset Statistics for each dataset and distribution shift scenario. Table 5 provides more fine-grained information that reflects the characteristics of the constructed datasets and distribution shifts. Table 6 details the domain splits and subgroup splits for each shift scenario.
>
> We thank the reviewer’s advice and will integrate these into a consolidated table for greater clarity in the refined manuscript.
> ### 1.2 More simulation and experimental details of the datasets
> Thank you for your suggestion. We have included basic information such as backgrounds, task definitions, and distribution shifts in Section 3.2 of the main text, with additional details provided in Appendix C. Additionally, Table 5 offers more fine-grained details, such as the average radius for each type of signal track, which helps readers grasp key characteristics of the proposed distribution shifts. We are happy to include more information if the reviewer is interested in specific details.
> ### 1.3 Motivations of using the selected GDL baseline models
> We have provided motivations for selecting the GDL models and learning algorithms in Appendix D of the initial manuscript. Please kindly check “Discussions” part of Appendix D1 and D2 (page 23-24) for details.
>
> ## 2. Additional Feedback
> ### 2.1 Domain-specific metrics for evaluation: For material science, are there any energy-based evaluations?
> We thank the reviewer for their valuable advice and agree that using domain-specific metrics is crucial for providing meaningful evaluations for domain practitioners.
>
> For material science, energy-related evaluations—such as tasks to predict formation energy or total energy (e.g., in the Materials Project dataset)—are indeed significant. However, to our best knowledge, none of these benchmarks currently offer multi-fidelity labels for the energies. This is why we selected the QMOF dataset, which involves predicting the band gap under fidelity shift, an important shift mechanism we aim to explore in this work, rather than others like the Materials Project.
>
> We appreciate the reviewer's suggestion and will consider incorporating more domain-specific evaluations in future work.
> ### 2.2 Do the authors observe the similar performance of OOD methods across different scientific datasets?
> Thank you for your question. Our answer is yes: some OOD methods do exhibit similar performance across various scientific scenarios. We provide three examples below:
>
> - **Case1**. Fine-tuning with quite limited OOD data (such as $\rm TL_{100}$ evaluated in this work) can result in negative effects compared to vanilla ERM across various scientific datasets， especially in cases involving a smaller degree of distribution shifts. Please refer to the related analysis in Sec 4.2 (Page 8).
> - **Case2**. The DIR method evaluated in this work yields inferior performance than vanilla ERM across various scientific fields. This is because the underlying assumptions are too rigid and fail to align with the shift mechanisms inherent in various proposed dataset. A similar analysis can be found in Point H1 of Appendix H (page 25).
> - **Case3**. As discussed in Point H4 of Appendix H (Page 26), TL methods show limited performance gains in both Assay shifts (Biochem dataset) and HSE06*-fidelity shifts (materials-science dataset). This could be attributed to the fact that both scenarios involve underlying mechanisms of concept shift with similar (small) degrees of shift in $P_{\mathcal{S}}(Y)$ and $P_{\mathcal{T}}(Y)$.
>
> Finally, we would also like to kindly note that given that each scientific dataset has its unique shift mechanism and varying degrees of shift intensity, it is generally challenging for a single method to perform equally well across all datasets.

---

> > ### Comment · Reviewer_UETZ · 2024-08-31
> >
> > Thanks for the rebuttal. The authors have addressed my concern. I maintain my score.

---

### Official Review · Reviewer_Ft87 · 2024-07-25
**Review for GeSS**

**Rating:** 6
**Confidence:** 3
**Correctness:** The claims in the submission are corr…
**Clarity:** The paper is well-written.

**Review:**

Quality and Clarity: The paper is well-written and structured. The methodology section details the experimental setup across 30 different settings, evaluating three GDL backbones and 11 learning algorithms.

Originality: The approach to creating a specialized benchmark for GDL in the context of distribution shifts is original and provides a valuable tool for future research in this area. This benchmark is unique not only for its focus on GDL but also for its comprehensive evaluation across multiple types of distribution shifts (conditional, covariate, and concept shifts) and various levels of OOD data access.

Significance: This paper is significant in that it can potentially drive forward the robust application of GDL in scientific domains where distribution shifts can affect predictive performance. The work provides a structured framework to assess and understand these shifts.

Pros:
- The experimental design is detailed, with specific attention to the types of distribution shifts and their implications in three different scientific domains.
- It provides extensive empirical evaluations, which reveal that no single approach excels across all types of shifts, highlighting the complexity of the problem.

Cons: While the paper discusses the benchmarks' theoretical aspects, it could further explore practical deployment challenges.

**Strengths:**

The review covers the main points. The study's context seems novel and valuable, and the empirical experiments are also comprehensive and detailed.

**Additional Feedback:**

There's no additional feedback.

**Documentation:**

The work is well-documented.

**Ethics:**

There is no ethical concerns.

**Limitations:**

The main limitation is that the experiments are designed to inform domain-specific practitioners. Still, given that GDL is not the only tool available, it is unclear how one might hold these results against others.

**Opportunities For Improvement:**

The authors could consider extending the comparison with existing benchmarks or methodologies outside GDL could strengthen the argument for the specific approaches taken.

**Relation To Prior Work:**

Section 2 of the paper discusses its relation to prior work.

**Summary And Contributions:**

The paper introduces GeSS, a comprehensive benchmark for evaluating the performance of Geometric Deep Learning (GDL) models under various scientific applications that experience distribution shifts. It addresses the lack of exploration in GDL's ability to manage distribution shifts by designing a framework that evaluates GDL models across diverse domains such as particle physics, materials science, and biochemistry, considering shifts like conditional, covariate, and concept shifts. The benchmark tests three types of information access scenarios regarding out-of-distribution (OOD) data, contributing to understanding how GDL can be optimized across different scientific fields under distribution shifts.

---

> ### Author Rebuttal · Authors · 2024-08-15
>
> We thank the reviewer Ft87 for recognizing the well-written paper, detailed experimental design, and extensive empirical evaluations of this work. We summarize the reviewer’s concerns and provide our explanations as follows. If your concerns are satisfactorily addressed, we would be grateful if you could consider improving the rating of our work.
> ## 1. Extending the comparison with existing benchmarks or methodologies outside GDL
> This section is to reply to the review quoted as `consider extending the comparison with existing benchmarks or methodologies outside GDL` and `Still, given that GDL is not the only tool available, it is unclear how one might hold these results against others.`
>
> We thank the reviewer for this insightful suggestion. Actually, some of our observations can be extended and be compared with findings outside GDL. We have conducted detailed comparisons in Appendix I (pages 26-27) of the initial manuscript, including four pieces of "**consistency**" (i.e., findings that are also applicable outside GDL) and four pieces of "**disparity**" (i.e., findings that are novel and specific to GDL) between findings of this work and those in existing benchmarks or methodologies outside GDL. We apologize if this was not clear due to the placement in the appendix. We will use LaTeX boxes to make these comparisons more prominently referenced in the final version.
> ## 2. Experiments are designed to inform domain-specific practitioners
> Thank you for your feedback. While this benchmark is indeed built upon datasets and distribution shifts with real-world scientific backgrounds, its relevance extends beyond domain-specific applications and this work can offer important insights and contributions to ML and GDL researchers. We provide explanations from the two aspects.
>
> 1. **Distribution shift formulation**: We have abstracted the mathematical formulations of these domain-specific shifts and categorized them from an ML perspective, employing the tool of causality, which is used in ML literature [1-4]. **This ensures that the established formulations are not limited to a specific domain but are applicable to broader ML research.** Particularly, the proposed $\mathcal{C}$-conditional shift in this work is relatively less studied in the ML community.
> 2. **Experimental Observations** also provide valuable insights to the ML community and inspire ML researchers. For example, we find that domain adaptation (DA) strategies excel in the $\mathcal{C}$-conditional shift, where the variation arises in the geometric characteristics of the causal component (see Conclusion 1 on page 8). We analyze these findings empirically from an ML viewpoint, and we believe that **these conclusions could be extended to general GDL and ML settings beyond scientific applications**.
>
> Thus, we assert that the benchmark and experiments are designed to not only inform domain-specific practitioners but also offer significant contributions to ML and GDL researchers.
> ## 3. Further exploration of practical deployment challenges
> We appreciate the reviewer’s suggestion to further explore practical deployment challenges and agree that addressing deployment challenges is important for real-world applicability.
> In the final paragraph of Appendix E of the initial manuscript (page 24, `Accordingly, …`), we have proposed **three steps** for GDL practitioners as a potential solution in handling distribution shift issues under practical scenarios, which could be seen as a potential exploration for practical deployment challenges.
>
> - Step1.  Assess the mechanism of shift by leveraging domain-specific knowledge;
> - Step2.  Assess the availability of collecting labeled or unlabeled OOD data;
> - Step3.  Select the suitable Info level and algorithms by the trade-off between the expenses of gathering extra OOD information and the expected performance gain resulting from such info.
>
> Based on the reviewer's suggestions, we plan to do the following for further exploration.
>
> - Firstly, in the final version, we will add a dedicated discussion section titled “Exploration of Practical Deployment Challenges” to highlight the proposed solutions.
> - In this new section, we will also identify potential challenges associated with deploying the solution in practice. For example, in step1, identifying the mechanism of the targeted shift may be difficult in highly intricate and time-varying environments. In step2, the assessment depends on the available computational resources. And in step3, determining the trade-offs requires empirical experience.
> - Looking ahead, we will collaborate with scientific experts and do evaluations in more practical deployment settings as future work to seek more insights and find more practical solutions.
>
> ## References
> [1] Invariance principle meets information bottleneck for out-of-distribution generalization. Advances in Neural Information Processing Systems, 34:3438–3450, 2021.
>
> [2] Invariant risk minimization. arXiv preprint arXiv:1907.02893, 2019.
>
> [3] Learning causally invariant representations for out-of-distribution generalization on graphs. Advances in Neural Information Processing Systems, 35:22131–22148, 2022.
>
> [4] Discovering invariant rationales for graph neural networks. In International Conference on Learning Representations, 2021.

---

> > ### Comment · Reviewer_Ft87 · 2024-08-31
> >
> > Thank you for addressing my concerns. I will keep my score.

---

### Official Review · Reviewer_yPRy · 2024-07-31

**Rating:** 6
**Confidence:** 3
**Correctness:** Yes
**Clarity:** Yes

**Review:**

Please refer to other parts

**Strengths:**

1.the benchmark would be very useful for the research community.

2.the experimental results are solid, and indeed shows useful take-away findings.

**Additional Feedback:**

No

**Documentation:**

Yes

**Ethics:**

No issues

**Limitations:**

Yes

**Opportunities For Improvement:**

1.the motivation of the benchmark is not clear, as it is unknown what is the most important contribution of the benchmark. I suggest the authors to reorganize the whole paper, and highlight the key contribution and designs of it.

**Relation To Prior Work:**

Yes

**Summary And Contributions:**

This work evaluates existing GDL models under the distribution shifts setting. The proposed benchmark has 30 distinct scenarios with 10 shift cases times 3 levels of available OOD info, covering 3 GDL backbones and 11 learning algorithms. The evaluation results have shown interesting findings.

---

> ### Author Rebuttal · Authors · 2024-08-15
>
> We thank the reviewer yPRy for recognizing the solid empirical evaluations and useful take-away findings of this work. We also thank the reviewer for the valuable feedback. We apologize for the potential confusion about the motivation / contribution statement and will improve our logic in the final version. Here we appreciate the opportunity to clarify the motivation behind this benchmark and to highlight its key design elements and contributions. Also, if your concerns are satisfactorily addressed, we would be grateful if you could consider improving the rating of our work.
> ## 1. Motivation Restatement (Para. 1-5 of Intro)
> Here we present the motivation logic by bullets as follows.
> 1. (Para. 1-3 of Intro) GDL plays a crucial role in various scientific fields. However, despite its importance, rare works have studied how to improve its ability to tackle the distribution shift problem, a prevalent challenge in many applications. **This point guided us to focus this benchmark on GDL under Scientific Applications with Distribution Shifts.**
> 2. (Para. 4 of Intro) Previous studies in ML4S often focus on specific scientific scenarios with specific distribution shifts, leading to findings that may be effective for one application but not applicable to others. **This limitation highlighted the need for a more comprehensive benchmark which covers various scientific scenarios and diverse shift mechanisms.**
> 3. (Para. 5 of Intro) Existing studies often assume different levels of availability of target-domain data, which can dedicate the choice of learning algorithms. However, how different availability of target domain data may impact the selection of methods has not been well studied. **This motivated us to consider different levels of target information availability simultaneously, to determine the most suitable approaches.**
> ## 2. Key Design Elements Restatement (Para. 6 of Intro)
> Inspired by the motivation above, here we present the Key Design Elements by bullets as follows.
> 1. **Diverse Coverage of Scientific Fields** (Inspired by Motivation 2): This benchmark covers various scientific fields with diverse mechanisms of distribution shifts. This diversity allows for a comprehensive evaluation of GDL generalizability under varying conditions.
> 2. **Three Distinct Information Levels** (Inspired by Motivation 3): For each case in the benchmark, we establish three distinct information levels: No-Info, O-Feature, and Par-Label. We evaluate corresponding learning algorithms at each level, enabling a co-analysis between shift mechanisms and information levels.
> ## 3. Contributions
> With all the motivations and key designs stated above, we want to highlight several most important contributions of this benchmark.
> 1. **Comprehensive benchmark**: Evaluate GDL generalizability over 3 scientific domains, 6 datasets, and 30 different settings with 10 distinct distribution shifts times 3 levels of OOD info, covering 3 GDL backbones and 11 learning algorithms in each setting.
> 2. **Theoretical formulation of distribution shifts**: Abstract the mathematical formulation of distribution shifts from scientific applications to support result analysis and extend the applicability of this benchmark beyond scientific applications, making it relevant for a broader range of scenarios.
> 3. **Experimental designs with practical motivations**: Consider multiple levels of target information availability in experiments to align with practical use cases. This design ensures the benchmark reflects practical challenges faced in deploying GDL models.
> 4. **Findings that help advance GDL research**: Reveal critical insights by highlighting the importance of understanding the interaction between distribution shift mechanisms and information levels. These findings not only guide the selection of appropriate algorithms but also contribute to the broader GDL research community.
>
> We will highlight key contributions more in the final version.

---

> > ### Comment · Reviewer_yPRy · 2024-08-30
> > **Response to Authors**
> >
> > Thank you for your clarification, and part of my concern has been addressed. I will keep my score as it is positive.

---

> > > ### Author Response · Authors · 2024-08-30
> > > **Thanks for the feedback**
> > >
> > > Many thanks for checking our response. Thank you for your feedback and support!
> > >
> > > the authors

---

### Author Rebuttal · Authors · 2024-08-15

# Rebuttal Summary
Dear reviewers,

Many thanks for your time and efforts in providing us with these valuable comments to improve the paper presentation. We are grateful that reviewers appreciate the efforts and contributions of our work in developing a new benchmark for geometric deep learning that spans various scientific applications and types of distribution shifts.

Also, we appreciate the actionable feedback from each reviewer for us to improve our work further. In the following, we will address their comments respectively in our response to each reviewer. If your concerns are well resolved, we would be grateful if you could consider enhancing the rating of our work.

Best,

Authors

---

### Author Response · Authors · 2024-08-23
**Looking forward to feedback from the reviewers**

Dear reviewers,

We would like to kindly draw your attention to our submission. After thoroughly considering your comments, we have provided point-by-point responses to address all your concerns and questions. As it is approaching the end of the discussion period, we would greatly appreciate any further feedback you might have, as it is crucial for the improvement of this work. Also, if you find that your concerns have been satisfactorily addressed, we would be grateful if you could consider improving the rating of our submission.
Thanks for your attention and participation!

Best regards,

The Authors

---

> ### Author Response · Authors · 2024-08-27
> **Looking forward to feedback from the reviewers**
>
> Dear reviewers,
>
> We would like to kindly draw your attention to our submission. After thoroughly considering your comments, we have provided point-by-point responses to address all your concerns and questions. As the discussion period is closing, we would greatly appreciate any further feedback you might have, as it is crucial for the improvement of this work. Thanks for your attention and participation!
>
> Best regards,
>
> The Authors

---

### Comment · Area_Chair_RJF4 · 2024-08-31
**Feedback on rebuttals**

Dear Reviewers, Thank you for your reviews. Please review the author rebuttals and give feedback as the deadline is tomorrow for the interactions. Thank you. Best, AC

---

### Decision · Program_Chairs · 2024-09-26

**Decision:**

Accept (Poster)

**Comment:**

Since both AC and SAC are non-responsive, PCs decide to give accept based on current reviewers' comments.